# AutoWeave: Automating Web Workflow Execution with Prompt-Adaptive Multi-Agent Orchestration

## Abstract

Performing tasks automatically over the web using LLM-based agents has witnessed a recent surge in interest. Executing a web task based on the intent expressed by a user requires carrying out a sequence of steps which presents several challenges owing to the complex nature of web workflows and variations across web interfaces. Prior works that have leveraged agentic framework for web workflow execution either employ a **fixed static call sequence** while invoking LLM agents or stack calls to **code-based functions** during runtime. Further, limited attention has been given to designing adaptable LLM-based web agents with **dynamically tunable prompts**. To this end, we propose AutoWeave, an agentic framework comprising of a suite of LLM-based agents to **anticipate future possibilities** due to an action by **looking-ahead** and simulate the suitability of actions during each step of workflow execution. The deliberation between the agents is facilitated by an **orchestrator LLM agent** which dynamically invokes the next appropriate agent based on interaction between the agents and the workflow executed so far. In addition, the orchestrator agent **refines the prompt** for each agent based on the task context before calling it during deliberation. We establish the efficacy of AutoWeave on a variety of benchmarks comprising 1) real-world websites like WebVoyager and 2) simulated web environments like WebArena with relative gains of **10%** and **22%** respectively over the best baselines. We show that AutoWeave consistently improves the performance of LLM-based web agents for multiple model families like Llama-3 and Qwen-2.5. Further, we conduct extensive ablations to verify the effectiveness of each agent in AutoWeave and the importance of Orchestrator for dynamic invocation of agents and prompt adaptation.

## 1 Introduction

Advancements in Large Language Models (LLMs) have paved the way for their use in developing autonomous agents (Gao et al., 2024; Xi et al., 2025; Patnaik et al., 2025) capable of executing complex workflows on web interfaces (Abuelsaad et al., 2024a; Zheng et al., 2024). To accomplish a given task based on user intent, such agents must interpret the web page using its Document-Object Model (DOM) (Nakano et al., 2021) or Accessibility Tree (He et al., 2024), and orchestrate a sequence of actions within the web browser. Nevertheless, autonomous execution of web workflows remains highly challenging owing to significant variation across web interfaces that causes the *observation* and the *action* spaces of web tasks to diverge from the parametric knowledge of LLM-based agents (Yang et al., 2025). Moreover, the DOM objects of a web page can be large, and noisy, making it difficult for LLM to focus on task-relevant elements as it is prone to distraction from irrelevant context (Liu et al., 2024). These issues get exacerbated for *long-horizon* web workflows requiring agents to handle substantial context accumulated during workflow execution steps.

Prior work has explored multiple strategies to improve agent performance on handling web-based tasks. One set of methods focuses on mitigating noise by filtering out irrelevant information from the web page's DOM (Lee et al., 2025; Yang et al., 2025). Agent-E (Abuelsaad et al., 2024a), for instance, employs a method that **programmatically** extracts the DOM elements required at a given step of workflow execution. While this reduces the observation size, the filtering is performed

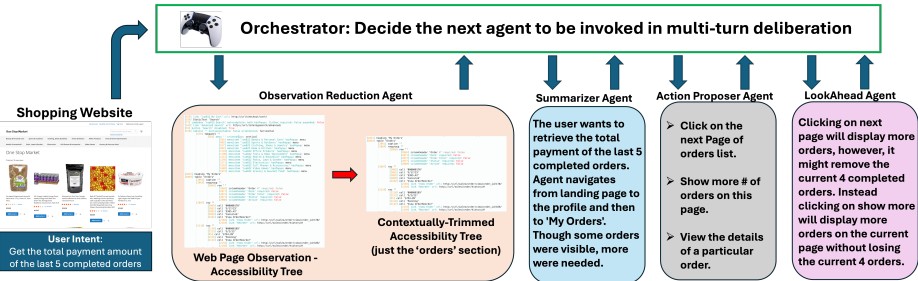

Figure 1: Overview of AutoWeave framework for web task automation. We conduct a multi-turn deliberation between diverse agents to decide the action for the current workflow step. The sequence and choice of agents to invoke during deliberation is decided by an orchestrator LLM agent.

heuristically by ignoring the surrounding context which could lead to loss of useful information. In our work, we employ an LLM-based agent that takes the entire accessibility tree as input to distil it and reduce its size conditioned on the **entire surrounding context of each element**. Furthermore, most existing frameworks rely on pre-defined, static sequences to determine the invocation order of various agents (Shinn et al., 2023). This rigid control flow limits the agent's ability to generalize and adapt to new observations that arise dynamically during the execution of web workflows. Although, SteP (Sodhi et al., 2024) addresses this to some extent by decomposing tasks into sub-tasks and dynamically invoking **code-based functions** (whose parameters are decided by an LLM) for each sub-task. However, SteP's scope is restricted to invoking code-level function control and it does not explore dynamic coordination between **LLM-based agents**. Also, while some recent work has explored designing **fixed hand-crafted prompts** for LLM-based agents (Fu et al., 2024), relatively little attention has been devoted to designing robust and adaptable LLM agents for web tasks with **dynamically adjusted and refined prompts** based on the context of workflow execution.

We define an agent to be an LLM which is instructed to perform a certain role and address the aforementioned limitations by investigating the following research questions (RQ) – **RQ1:** *Can deliberation among LLM agents designed to simulate the suitability of actions through **prompts which can be tuned dynamically by another agent** – lead to more effective execution of web tasks?*; and **RQ2:** *Does the **dynamic orchestration of these LLM agents** during deliberation, informed by the state of inter-agent discussions and the evolving workflow context, enhance overall performance?* Motivated by these research questions, we introduce **AutoWeave**, a framework that **1)** conducts deliberation among LLM agents to simulate the potential future states and web pages which can result from a given action, and selects the next best action based on this simulation at each step of workflow execution; **2)** employs a dedicated *orchestrator agent* that dynamically decides the LLM agent to be invoked at each turn of deliberation based on the workflow context (as shown in Figure 1); and **3)** uses the orchestrator agent to refine the prompt of the invoked agent by enriching it with additional information and guidelines based on the deliberation and workflow context.

In particular, AutoWeave comprises of a *LookAhead* agent (§ 3.1) which judges the suitability of an action by forecasting possible future trajectories of the workflow that the action could lead to. It does so by anticipating the subsequent web pages that might come up and their alignment with the workflow goal. The *Orchestrator Agent* (§ 3.2) in AutoWeave dynamically decides the agent to be invoked at each turn of deliberation. Notably, the orchestrator adaptively refines the prompt of the selected agent with key deliberation highlights and possible suggestions on what to focus on, thus, enabling more context-aware reasoning aligned with the evolving workflow and the ongoing inter-agent discussion. AutoWeave also employs an LLM-based Observation Reduction agent that selectively retains relevant segments of the HTML tree by filtering out irrelevant elements that can act as noise to other agents. AutoWeave reduces the HTML tree conditioned on entire HTML tree as well as the workflow context so that the tree is reduced in a goal-aligned manner. Other works like Agent-E (Abuelsaad et al., 2024a) reduces the observation, however, they do not consider entire context of the HTML. similarly, AgentOccam (Yang et al., 2025) distills the HTML tree in isolation, disregarding both the workflow context and the task objective.

Empirically, we evaluate AutoWeave on benchmarks spanning real-world websites and simulated web environments (§ 4). Specifically, we evaluate on 1) WebVoyager (He et al., 2024), which in-

cludes a wide range of tasks drawn from 15 real websites such as Amazon, GitHub, and Google; and 2) WebArena (Zhou et al., 2024) which features simulated environments mirroring complex web interfaces like CMS, maps, and social forums. AutoWeave consistently outperforms several baselines (§ 4.1) achieving relative gains of 10% on WebVoyager and 22% on WebArena. Through comprehensive ablations (§ 4.3), we establish the efficacy of each agent and establish that *dynamic agent orchestration with prompt-refinement*, enabled by the LLM-based orchestrator agent, is a key driver of AutoWeave's outperformance compared to task-agnostic static agent-invocation graph (§ 4.3). Finally, we show that AutoWeave generalizes across model families demonstrating strong performance on Llama-3-70B/405B and Qwen-2.5-72B (§ 4.2). Since multi-turn deliberation between LLM agents at each web workflow execution step results in large token overheads, we do not use GPT-4 in experiments owing to high API costs. In contrast, open-source models such as Llama-3 and Qwen-2.5 can be GPU-hosted without token costs, thus, enabling large-scale experiments.

## 2 RELATED WORK

**LLMs as Execution Agents:** Early works on instruction-based task execution focused on mapping natural language instructions to corresponding actions (Branavan et al., 2009; Artzi & Zettlemoyer, 2013; Liu et al., 2018; Humphreys et al., 2022). The emergence of LLMs with strong reasoning capabilities (Wei et al., 2022; Chu et al., 2024) enabled their use to drive planning in software and embodied environments (Baker et al., 2022; Wang et al., 2024a;b; Ahn et al., 2022; Bousmalis et al., 2024; Wu et al., 2024; Bhateja et al., 2023). Frameworks such as Reflexion (Shinn et al., 2023) and React (Yao et al., 2023) have demonstrated the use of LLMs as agents to design inference-time execution strategies. This has spurred the development of LLM-based web agents which could be uni-modal (using just the HTML representation of web page) (Sodhi et al., 2024; Yang et al., 2025) or multi-modal (leveraging the image of the web page) (Zheng et al., 2024; He et al., 2024). In this work, we focus on text-only unimodal LLM agents, leaving multimodal extensions for future work.

**LLM Agents for Web Tasks:** Recent approaches have emphasized planning and decomposition to handle complex web tasks by breaking them down into simpler subtasks (Sun et al., 2023; Prasad et al., 2024; Erdogan et al., 2025). Plans can also be improved based on feedback from the environment (Sun et al., 2023) depending on whether the actions taken advanced the workflow (Lutz et al., 2024; Pan et al., 2024). In contrast, AutoWeave leverages agents to *proactively* deliberate and evaluate the suitability of actions *a priori*, rather than relying on post-execution feedback. Several approaches rely on fine-tuning LLMs for web navigation (Deng et al., 2023; Furuta et al., 2024; Yao et al., 2022; Yin et al., 2024; Hong et al., 2024; Lai et al., 2024; Putta et al., 2024), but these are constrained by the scarcity of annotated long-horizon workflows. Attempts have been made to address data limitations by leveraging past experiences (Fu et al., 2024; Wang et al., 2024c; Zheng et al., 2023; Kagaya et al., 2024) or online tutorials (Ou et al., 2024) through in-context learning (Zhou et al., 2024; Kim et al., 2023). However, these suffer from poor generalization due to the diversity in web interfaces, and they often result in a very large input context degrading LLM performance. Additional related work is discussed in Appendix A.1.

**Decision-Making for Agent Orchestration:** LLMs have shown remarkable decision-making capabilities (Huang et al., 2022; Brown et al., 2020) for chaining reasons and actions (Schick et al., 2023; Yang et al., 2023). However, most prior methods manually define a static graph to decide order of invocation of APIs and agents. Reflexion (Shinn et al., 2023) comprises an actor agent to generate and self-reflect on the trajectory conditioned on feedback from an evaluator agent. Methods like DecomP (Khot et al., 2023) and SteP (Sodhi et al., 2024) decomposes a task into a static program to call subroutines. However, such works have not explored dynamic stacking of LLM-based agents.

## 3 METHODOLOGY

**Problem Formulation:** Given a natural language instruction $\mathcal{I}$ describing the task $\mathcal{T}$, the goal is to execute the task in the web interface by performing a workflow i.e. a sequence of actions $\{a_1, a_2, ..., a_t, ..., a_M\}$, where, $M$ is the maximum number of steps. At each time step $t$, to advance the workflow towards task completion, AutoWeave is required to decide the action $a_t$ to be performed based on the current observation $o_t$ of the environment and the history $h_t = \{a_{t-1}, o_{t-1}, ...\}$ of past workflow steps. The current observation $o_t$ is the accessibility tree of the current web-page

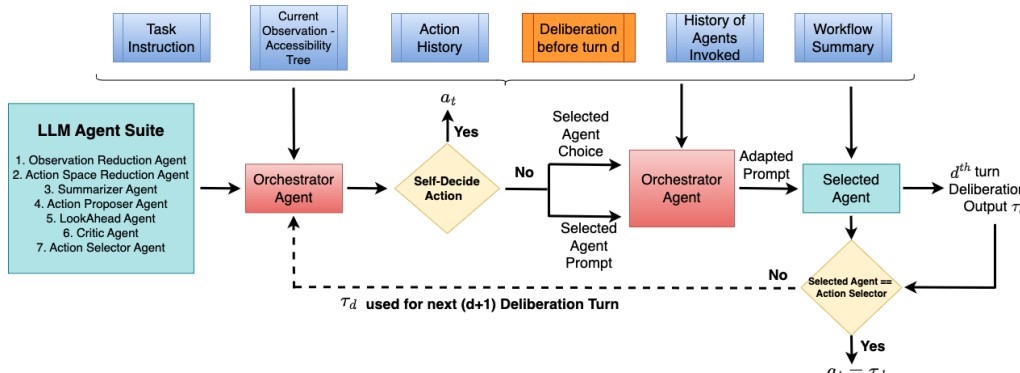

Figure 2: Schematic view depicting a deliberation turn $d$ while determining the action at workflow step $t$. The orchestrator decides whether to self-infer the action or invoke the next agent. In the latter case, it adapts next agent's prompt such that the agent output is used for next turn of deliberation.

HTML serialized as text. The observation $o_t$ is a partial view of the state $s_t$ of the environment since the full state comprises other aspects such as data. Taking action $a_t$ changes the state of the environment from $s_t$ to $s_{t+1}$. The actions $a_t \in \mathcal{A}$, such that $\mathcal{A}$ is the space of possible actions that can be performed on the webpage such as click[id], type[id], etc. where 'id' refers to the element on the web page using which the action needs to be executed. We now discuss the details of AutoWeave.

**Overview of AutoWeave:** To infer the action $a_t$ to be executed at $t^{th}$ workflow step, AutoWeave conducts a multi-turn deliberation $\mathcal{D}_t = \{\tau_1, ..., \tau_d, ..., \tau_M\}$ between diverse LLM agents (as shown in figure 2). Here, M is the maximum number of deliberation turns. Each agent in the suite $\mathcal{LA}$ of LLM agents is defined to perform a specific role (§ 3.1). Specifically, at each turn $\tau_d$ during deliberation $\mathcal{D}_t$, an LLM-agent $\mathcal{LA}_d$ is invoked to perform its role conditioned on the workflow context ($o_t$ and $h_t$) as well as deliberation till turn $d$ ($\mathcal{D}_t^{<d}$). The decision of agent to invoke is made by an orchestrator (§ 3.2). Since LLMs have shown strong decision-making capabilities as well as the ability to chain reasons and actions, we use an LLM agent as orchestrator. The orchestrator agent $\mathcal{LA}^O$ selects the next agent $\mathcal{LA}_d$ conditioned on deliberation and workflow context and gives the control to it. $\mathcal{LA}_d$ then performs its role and returns the control to $\mathcal{LA}^O$. Further, $\mathcal{LA}^O$ assists $\mathcal{LA}_d$ before invoking it by adapting its prompt with highlights from deliberations till turn $d$ and suggestions on what to focus on. Once the orchestrator concludes the deliberation, it infers the action $a_t$ either by itself or invoke an action selector agent. We now discuss details of each component.

## 3.1 LLM Agent Suite in AutoWeave

We describe the suite of diverse LLM agents $\mathcal{LA}$ used in AutoWeave. Each LLM agent is defined through a prompt (please refer Appendix A.5 for prompt details) specifying the role of the agent.

1. **Observation Reduction Agent ($\mathcal{LA}^{OR}$):** Bulky nature of Accessibility Tree (AX Tree) results in very long inputs for LLMs making them error-prone. At a given workflow step, only a sub-part of the AX tree is relevant to the user task. Hence, we instruct the LLM to contextually identify the content, sections and elements in the AX tree which are useful to achieve the user goal and filter out the remaining irrelevant parts while maintaining the logical structure of the tree. Formally, $o_t^{'} = \mathcal{LA}^{OR}(o_t; h_t; \mathcal{I}; \mathcal{D}_t^{<d})$, such that $o_t^{'}$ represents contextually trimmed AX tree.

2. **Action Space Reduction Agent ($\mathcal{LA}^{ASR}$):** Certain actions can be extraneous to the current workflow step depending on the workflow context, current web page and user goal. For instance, 'fill' may not be required on a web page where there are no form fields. Thus, such actions can be removed from the action space to obtain a reduced set of actions $\mathcal{A}^{'}$ for simplifying action selection. This can be expressed through following equation: $\mathcal{A}^{'} = \mathcal{LA}^{ASR}(o_t; h_t; \mathcal{I}; \mathcal{A}; \mathcal{D}_t^{<d})$

3. **Summarizer Agent ($\mathcal{LA}^S$):** Due to long-horizon nature of web tasks, it is infeasible to use AX trees of all previous web pages in input owing to limited context window of LLMs and their vulnerability to long noisy-context. To use the workflow information to infer the next action in a

contextually-aware manner, we instruct the LLM to distill observation-action pairs history from last k steps into a summary $s$ capturing a gist of trajectory and changes in web pages' state. Thus, $s = \mathcal{LA}^S(\{o_{t-1}, a_{t-1}, ..., o_{t-k}, a_{t-k}\}; \mathcal{I}; \mathcal{D}_t^{<d})$ serves as workflow context for different agents.

4. **Action Proposer Agent ($\mathcal{LA}^{AP}$):** Proposes a set of action candidates $\mathcal{PA}$ which could possibly help advancing the workflow based on task instruction, current AX tree, history of actions, context summary and deliberation $\mathcal{D}_t^{<d}$. $\mathcal{PA} = \mathcal{LA}^{AP}(o_t; s; \mathcal{I}; \mathcal{A}; \mathcal{D}_t^{<d}; \{a_{t-1}, ..., a_1\})$.

5. **LookAhead Agent ($\mathcal{LA}^{LA}$):** Given an action candidate $a$, judge its suitability by simulating future possibilities i.e. subsequent web pages along with corresponding elements and actions. The lookahead output description $\mathcal{LAD} = \mathcal{LA}^{LA}(a; o_t; s; \mathcal{I}; \mathcal{D}_t^{<d}; \mathcal{A}; \{a_{t-1}, ..., a_1\})$.

6. **Critic Agent ($\mathcal{LA}^C$):** Given an action candidate $a$, the LLM is instructed to play the role of an adversary and identify potential pitfalls and failure cases that might arise due to taking the action. The critic output description $\mathcal{CD} = \mathcal{LA}^C(a; o_t; s; \mathcal{I}; \mathcal{D}_t^{<d}; \mathcal{A}; \{a_{t-1}, ..., a_1\})$.

7. **Action Selector Agent ($\mathcal{LA}^{AS}$):** Once the deliberation concludes, the role of the action selector agent is to finally select the action to be taken based on entire deliberation context. The action $a_t$ chosen by action selector $\mathcal{LA}^{AS}$ can be written as, $a_t = \mathcal{LA}^{AS}(o_t; s; \mathcal{I}; \mathcal{D}_t^{<d}; \mathcal{A}; \{a_{t-1}, ..., a_1\})$.

We generically refer to the output of an agent at the $d^{th}$ turn as $\tau_d$. It should be noted that $\mathcal{D}_t^{<d} = \{\tau_{d-1}, ..., \tau_1\}$ refers to the deliberation till $d^{th}$ turn. Further, the workflow context summary $s$ is used as input for subsequent agents during deliberation if the summarizer agent is invoked (or else $s$ is omitted otherwise). Also, the candidate actions for lookahead, critic and action selector agents are decided based on the output of the action proposal agent. The decision regarding which agent to invoke when during deliberation is taken by an orchestrator which we discuss in the next subsection.

## 3.2 Orchestrator for Dynamic Invocation of LLM Agents

To infer the action $a_t$ at step $t$, AutoWeave conducts a multi-turn deliberation between the LLM agents. The choice of agent to be invoked at each turn d of deliberation depends on the task instruction, context of the workflow executed so far as well as the deliberation done till turn $d$. For instance, it might not be required to invoke the summarizer agent during the initial steps of the workflow. Likewise, the difficulty of deciding action for the current workflow step would influence the number of discussion turns and choice of action proposal, lookahead and critic agents. Hence, the decision to select the next agent during deliberation should be made dynamically to make the inter-agent interaction adapt to the task complexity as new observations arise during the workflow. Owing to the fact that LLMs have shown strong decision-making abilities, we use an LLM-based orchestrator agent $\mathcal{LA}^O$ to dynamically make the choice of next agent to be invoked at each deliberation turn. More formally, the output $\mathcal{O}^O$ of the orchestrator is obtained based on the task instruction $\mathcal{I}$, current web page observation $o_t$, summarized workflow context $s$ (if summarizer agent is invoked till turn $d$), deliberation $\mathcal{D}_t^{<d}$ till turn $d$, role definition ($\mathcal{RD}$) of each LLM agent, history of agents $\{\mathcal{LA}_{d-1}, .., \mathcal{LA}_1\}$ invoked, action space, and history of actions, as shown in the following equation:

$$\mathcal{O}^O = \mathcal{LA}^O(o_t; s; \mathcal{I}; \mathcal{D}_t^{<d}; \mathcal{RD}; \{\mathcal{LA}_{d-1}, ..., \mathcal{LA}_1\}; \mathcal{A}; \{a_{t-1}, ..., a_1\})$$

When orchestrator concludes the deliberation, it would either hand the control to the action selector agent or decide the action $a_t$ itself. Hence, the orchestrator output $\mathcal{O}^O$ is either the next agent $\mathcal{LA}_d$ or the action $a_t$ to be executed at workflow step t. In addition, the observation reduction ($\mathcal{LA}^{OR}$) and action space reduction ($\mathcal{LA}^{ASR}$) agents are optionally invoked only at first and second turns of deliberation respectively. The orchestrator $\mathcal{LA}^O$ decides whether to call ($\mathcal{LA}^{OR}$) at the first turn. If yes, then contextually trimmed AX tree $o_t'$ is used for remaining deliberation. Likewise, if $\mathcal{LA}^O$ decides to invoke ($\mathcal{LA}^{ASR}$), then the reduced action space $\mathcal{A}'$ is used for subsequent deliberation.

**Prompt-Adaptive Agent Invocation:** To enable the agent $\mathcal{LA}_d$ (being invoked at turn $d$) adapt to the deliberation and workflow context, the orchestrator $\mathcal{LA}^O$ is instructed to refine the prompt $\mathcal{P}_{\mathcal{LA}_d}$ of $\mathcal{LA}_d$ before calling the agent. $\mathcal{LA}^O$ contextualises $\mathcal{P}_{\mathcal{LA}_d}$ with key highlights from the deliberation and suggestions about the aspects that can be focussed by $\mathcal{LA}_d$. Thus, the modified prompt $\mathcal{P}_{\mathcal{LA}_d}' = \mathcal{LA}^O(\mathcal{I}; o_t; \mathcal{A}; \{a_{t-1}, ..., a_1\}; s; \{\mathcal{LA}_{d-1}, ..., \mathcal{LA}_1\}; \mathcal{D}_t^{<d})$. We now discuss the experiments conducted to evaluate the efficacy of AutoWeave and its different components.

Table 1: AutoWeave outperforms all the baselines on the 6 WebArena environments (22.1% relative outperformance over the best baseline). All methods use Llama3-70B-instruct as the base LLM.

| Method | Shopping (#192) | CMS (#182) | GitLab (#196) | Map (#112) | Reddit (#114) | Wikipedia (#16) | Overall (#812) |
|---|---|---|---|---|---|---|---|
| Llama3 70b Instruct | 13.27 | 15.73 | 20.37 | 10.25 | 14.84 | 12.5 | 15.28 |
| BrowserGym | 16.49 | 22.94 | 22.73 | 13.29 | 16.17 | 6.25 | 16.40 |
| Tree Search | 21.35 | 25.27 | 30.10 | 17.50 | 15.28 | 18.75 | 22.91 |
| STeP | 30.84 | 28.13 | 22.71 | 23.93 | 25.17 | 31.25 | 26.48 |
| AgentOccam | 24.03 | 27.20 | 26.71 | 20.91 | 22.62 | 12.5 | 24.51 |
| Agent-E | 27.79 | 25.51 | 33.20 | 25.26 | 23.18 | 25 | 27.34 |
| **AutoWeave** | **35.97** | **30.72** | **36.28** | **31.72** | **35.48** | **43.75** | **33.37**$_{22.1\uparrow}$ |

## 4 EXPERIMENTS AND EVALUATION

**Benchmarks and Evaluation Metric:** We evaluate AutoWeave on two benchmarks: 1) WebVoyager (He et al., 2024), with **629** tasks from **15** real-world websites such as Google, Amazon, and GitHub, requiring navigation, dynamic content interaction, form filling, etc., and serving as a *real-world testing interface* for assessing generalizability; and 2) WebArena (Zhou et al., 2024), with **812** tasks across **6** simulated environments spanning ecommerce (OneStopShop), social forums (Reddit), DevOps (GitLab), and content management (online store management), as well as maps and Wikipedia, testing agents' ability to reason over structured and unstructured content, perform search, filter data, and more. We report average task success rates on both benchmarks. For WebVoyager, evaluation is done by prompting GPT-4o to judge whether the final page state after the actions satisfies the task's goal condition. For WebArena, a task is considered successful if the agent's final output matches the ground-truth answer via exact string match or GPT-4o–based "fuzzy" match. Further details about the evaluation metrics are provided in Appendix A.2.

**Baselines:** We compare AutoWeave with a comprehensive set of recent baselines: (1) BrowserGym (de Chezelles et al., 2025), the WebArena method using an agent with CoT prompting; (2) a Tree Search method (Koh et al., 2024b) performing best-first inference-time search over an LLM agent's actions for multi-step web planning; (3) SteP (Sodhi et al., 2024), which uses "stacked" LLM subroutine calls to decompose complex web tasks; (4) AgentOccam (Yang et al., 2025), a simple LLM-based agent aligning observations and actions with the LLM's pre-training; and (5) Agent-E (Abuelsaad et al., 2024b), a hierarchical web agent using DOM denoising and change detection to navigate complex tasks. We also include the underlying instruct version of the LLM (referred to as the 'LLM-Instruct' baseline). For WebVoyager, we additionally evaluate the text-only method proposed by (He et al., 2024), an LLM-based agent interacting end-to-end with real websites.

**Implementation Details:** HTML web page is represented using its AX tree. Decoding uses a temperature of 0.4 and a beam size of 3. We use 8 parallel workers for efficient task execution. Maximum number of deliberation turns ($N$) is 10, and maximum workflow steps ($M$) is 30.

### 4.1 DOES AUTOWEAVE PERFORM BETTER THAN BASELINES?

Tables 1 and 2 summarize the results on WebArena and WebVoyager respectively (using Llama3-70B-instruct as underlying LLM) where it is observed that AutoWeave provides substantial improvements over existing baselines. On WebArena (Table 1), AutoWeave achieves an overall success rate of 33.37%, outperforming the best baseline i.e. Agent-E by 6.03% (relative gain of 22%). Notably, AutoWeave shows significant gains in domains such as Shopping (+8.18% vs. Agent-E and +5.13% vs. best-baseline on Shopping i.e. STeP), Map (+6.46% vs. Agent-E), Reddit (+12.3% vs. Agent-E and +10.31% vs. STeP) and Wikipedia (+18.75% vs. Agent-E and +12.5% vs. STeP). On WebVoyager (Table 2), AutoWeave outperforms previous best method (Agent-E) by +4.3% on overall success rate (relative gain of +10.01%), with remarkable improvements on sites like Amazon (+5.1% vs. Agent-E), Google Flights (+4.0%), and GitHub (+5.5%).

Figure 3 shows a qualitative example where the user requests '*total amount of last 5 completed orders*' on a shopping website. AutoWeave navigates to '*My Orders*' page where the orchestra-

Table 2: AutoWeave outperforms all the baselines on the 15 WebVoyager environments (10% relative outperformance over the best baseline). All methods use Llama3-70B-instruct as the base LLM.

| Method | All Recipes | Amazon | Apple | ArXiv | Github | Booking | ESPN | Coursera |
|---|---|---|---|---|---|---|---|---|
| Llama3 70b Instruct | 19.23 | 22.45 | 25.13 | 24.10 | 20.36 | 18.75 | 21.45 | 23.98 |
| WebVoyager | 21.56 | 24.10 | 27.48 | 26.02 | 22.37 | 20.52 | 23.34 | 25.67 |
| BrowserGym | 25.67 | 28.39 | 30.45 | 28.76 | 26.40 | 23.10 | 26.72 | 27.98 |
| Tree Search | 28.12 | 30.67 | 32.91 | 30.45 | 28.33 | 25.20 | 28.94 | 30.65 |
| STeP | 30.34 | 32.45 | 34.89 | 32.78 | 30.56 | 27.35 | 31.67 | 33.45 |
| AgentOccam | 33.67 | 35.23 | 37.41 | 35.11 | 33.72 | 30.15 | 35.04 | 36.72 |
| Agent-E | 35.89 | 37.65 | 40.23 | 38.14 | 36.09 | 32.80 | 38.45 | 39.58 |
| **AutoWeave** | **40.23** | **42.75** | **44.67** | **43.12** | **41.56** | **38.90** | **43.78** | **45.11** |

| | Cambridge Dictionary | BBC News | Google Flights | Google Map | Google Search | HuggingFace | Wolfram Alpha | Overall |
|---|---|---|---|---|---|---|---|---|
| Llama3 70b Instruct | 24.78 | 22.39 | 26.45 | 23.71 | 25.89 | 27.34 | 28.56 | 26.23 |
| WebVoyager | 26.12 | 24.05 | 28.10 | 25.45 | 27.34 | 29.02 | 30.67 | 27.98 |
| BrowserGym | 28.45 | 26.72 | 30.11 | 27.34 | 29.67 | 31.23 | 33.10 | 31 |
| Tree Search | 31.23 | 28.89 | 33.45 | 30.78 | 32.56 | 34.12 | 36.34 | 35.39 |
| STeP | 34.10 | 31.67 | 36.78 | 34.12 | 35.67 | 37.45 | 39.56 | 37.52 |
| AgentOccam | 37.45 | 34.56 | 40.12 | 37.89 | 39.23 | 41.10 | 43.21 | 41.97 |
| Agent-E | 39.78 | 37.12 | 43.45 | 41.23 | 42.78 | 44.10 | 46.34 | 42.92 |
| **AutoWeave** | **44.12** | **41.23** | **47.45** | **45.67** | **46.78** | **48.56** | **50.34** | **47.22**$_{10.0\uparrow}$ |

Table 3: Evaluation of AutoWeave using LLMs from different model families and sizes (Llama-3 and Qwen-2.5). AutoWeave outperforms the best baseline i.e. Agent-E for both the LLMs showing its ability to generalize across LLMs with different underlying architectures and knowledge.

| Method | Model | WebArena | WebVoyager |
|---|---|---|---|
| Agent-E | Llama3-70b-Instruct | 27.34 | 42.92 |
| **AutoWeave** | Llama3-70b-Instruct | **33.37** | **47.22** |
| Agent-E | Qwen2.5-72b-Instruct | 26.41 | 40.43 |
| **AutoWeave** | Qwen2.5-72b-Instruct | **29.56** | **44.60** |
| Agent-E | Llama3.1-405b-Instruct | 39.19 | 52.08 |
| **AutoWeave** | Llama3.1-405b-Instruct | **45.22** | **56.17** |

tor initiates a deliberation between agents and invokes the action proposer agent which provides 3 candidate actions - clicking on *next page*; *show more*; or *view order details*. The orchestrator then invokes the lookahead agent and updates the prompt with the information that 4 orders are visible in current view. The lookahead agent then simulates the possibility that clicking on *next page* might result in loss of visibility of current 4 orders, and hence, favors clicking on *show more* which will likely show the 5th order. The baseline methods, instead chose to click on *next page* and got stuck in a loop. This demonstrates effectiveness of orchestrator-based invocation of agents with prompt-refinement. Please refer to Appendix A.6 for failure cases and qualitative analysis. Appendix A.3.2 shows how AutoWeave mitigates cyclic loops more effectively than baseline methods.

## 4.2 DOES AUTOWEAVE GENERALIZE ACROSS LLM FAMILIES AND SIZES?

Table 3 demonstrates versatility of AutoWeave across different model families. Specifically, we compare AutoWeave and the best-performing baseline Agent-E using LLMs from three distinct model families and sizes - Llama3-70b-Instruct, Llama3.1-405b-Instruct and Qwen2.5-72b-Instruct. Our approach consistently outperforms Agent-E across models on both WebArena and WebVoyager. When utilizing Llama3-70b-Instruct, AutoWeave achieved a WebArena success rate of 33.37%, surpassing Agent-E's 27.34%. Likewise, for WebVoyager, AutoWeave achieves a success rate of 47.22%, compared to Agent-E's 42.92%. A similar trend is observed with Qwen2.5-72b-Instruct and Llama3.1-405b-Instruct, where AutoWeave achieved a WebArena score of 29.56 and 45.22, outperforming Agent-E's 26.41 and 39.19. The performance gap is also significant for WebVoyager.

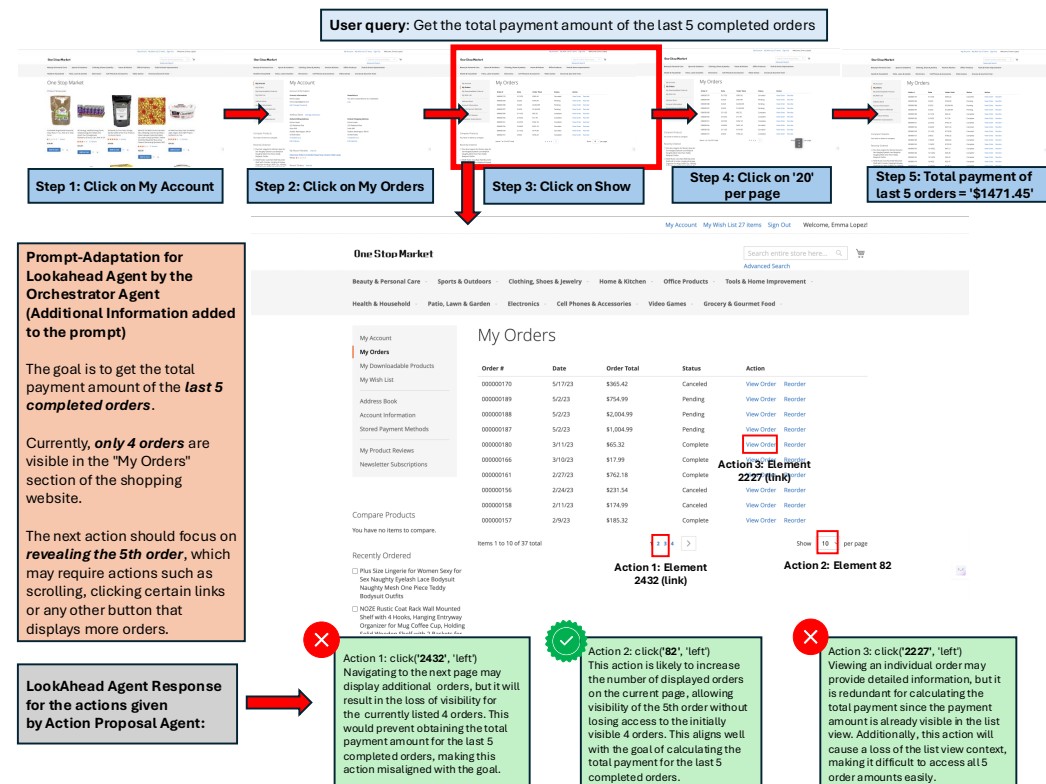

Figure 3: Qualitative example showing effectiveness of orchestrator-based agent invocation and prompt-adaptation. The user requests 'total amount of last 5 orders'. AutoWeave is able to complete workflow because at step 3, orchestrator refines lookahead agent's prompt with information - '4 orders are currently visible'. The lookahead agent simulates that clicking on show more will display more orders on current page itself. The baselines instead click on next page and gets stuck in a loop.

Table 4: Ablation study to validate the importance of each agent in AutoWeave . Performance drop in absence of agents like LookAhead, Critic, & Observation Reduction depict their importance. Dynamic invocation of agents using Orchestrator with prompt-adaptation is also crucial.

| Design Choice | WebArena | WebVoyager |
|---|---|---|
| **AutoWeave** | **33.37** | **47.22** |
| w/o Observation Reduction Agent | 29.06 | 42.13 |
| w/o Action Space Reduction Agent | 32.14 | 45.31 |
| w/o Summarizer Agent | 30.29 | 43.88 |
| w/o Lookahead Agent | 26.85 | 37.20 |
| w/o Critic Agent | 26.11 | 37.84 |
| w/o Selector Agent | 31.28 | 44.19 |
| w/o Orchestrator Agent for Dynamic Invocation | 27.34 | 42.77 |
| w/o Prompt-Adaptation | 25.37 | 39.58 |

## 4.3 UNDERSTANDING THE ROLE OF INDIVIDUAL AGENTS IN AUTOWEAVE

We conduct an ablation study to evaluate impact of each agent in AutoWeave. Table 4 depicts that AutoWeave with all the agents achieves the highest performance. Notably, the absence of the **Lookahead Agent** leads to the most significant drop to **26.85%** (↓6.52%) on WebArena and **37.20%** (↓10.02%) on WebVoyager, indicating that simulating the impact of actions is important. Omitting **Critic Agent** also have a substantial impact, reducing performance to **26.11%** (↓7.26%) on We-

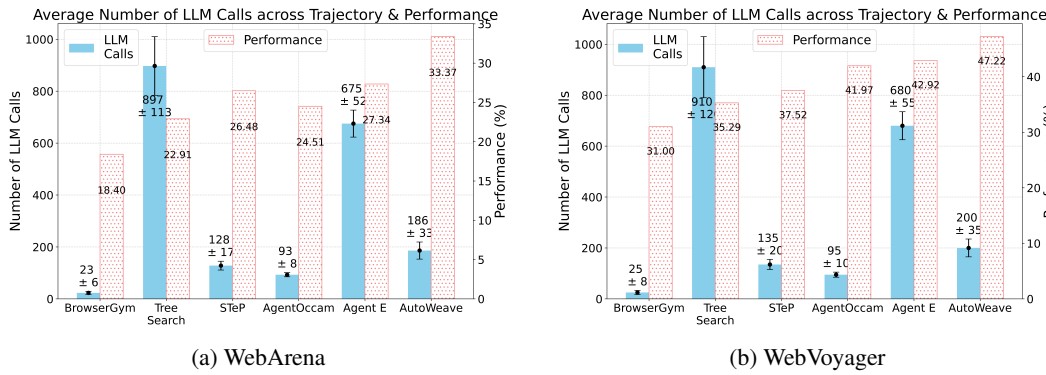

(a) WebArena

(b) WebVoyager

Figure 4: Comparison of accuracy-efficiency trade-off between AutoWeave and the baselines.

bArena and **37.84** ($\downarrow$9.38%) on WebVoyager. Omission of **Observation Reduction Agent** leads to drop of $4.31\%$ on WebArena and $5.09\%$ on WebVoyager. Dynamic invocation of agents by **Orchestrator** is critical for performance gains achieved by AutoWeave (drop of 6.03% on WebArena and 4.45% on WebVoyager in its absence). Disabling prompt-adaptation by the orchestrator leads to 7.99% drop on WebArena and 7.64% on WebVoyager, underscoring the value of prompt-adaptation. Appendix A.4 examines merging agent roles, which results in performance degradation.

### 4.4 ANALYSIS OF DELIBERATION TURNS, PERFORMANCE AND LATENCY TRADE-OFFS

Figures 4 and 5 summarize the efficiency and performance of AutoWeave. It attains the highest success rates (33.37% on WebArena, 47.22% on WebVoyager) with fewer LLM calls (186 ± 33, 186 ± 35) than Tree Search (897 ± 113, 897 ± 120) and Agent-E (675 ± 52, 675 ± 55). Agent-E, despite fewer agents, allows 30 deliberation turns and up to 30 steps per trajectory, often leading to loops and inflated calls. Dynamic orchestration in AutoWeave avoids redundant calls unlike exhaustive Tree Search.

Increasing deliberation turns (5, 10, 15) further boosts success (e.g., WebVoyager: 36%$\rightarrow$49%) but raises latency (75s$\rightarrow$160s). Gains are largest from 5 to 10 turns, with diminishing returns beyond, especially in WebArena. Overall,

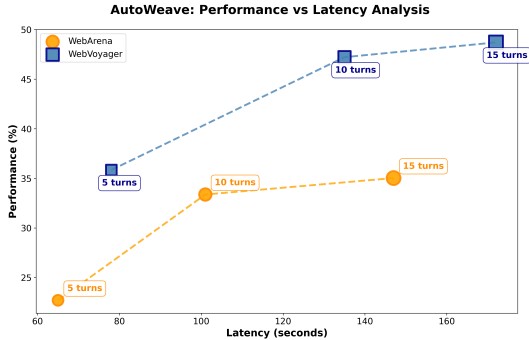

Figure 5: Performance vs Latency Trade-off

AutoWeave balances two key trade-offs: (i) accuracy vs. LLM calls, and (ii) accuracy vs. latency, offering tunability for deployment requirements.

**Limitations:** Limitations: While AutoWeave improves web task performance by leveraging comprehensive LLM agents, the diverse roles rely solely on the LLM's background knowledge, limiting adaptation to domain-specific workflows. Future work could explore using high-quality AutoWeave deliberations as training data to fine-tune models for better domain-specific generalization.

## 5 CONCLUSION

Prior works have given limited attention towards designing web agents which think about suitability of actions by looking-ahead the implications of an action. Further, most previous methods define a static call graph to decide order of invocation of agents which might not generalize well to all tasks. In this work, we propose AutoWeave, a framework comprising a suite of diverse LLM agents along with an orchestrator agent which conducts a deliberation between agents and dynamically makes the decision about which agent to invoke next during deliberation. We show that AutoWeave performs better than several baselines and generalizes well for LLMs from different model families.

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

## A APPENDIX

### A.1 ADDITIONAL RELATED WORK

Monte Carlo Tree Search (MCTS) has been used to perform roll-outs on web interfaces and obtain a value function for actions (Zhou et al., 2023; Zhang et al., 2025; Koh et al., 2024a). Similarly,

Chae et al. (2025) train a world model to simulate webpage transitions to learn an action policy. In contrast, AutoWeave employs a *lookahead agent* to simulate the possibilities likely to occur due to an action to assess its fitness. Some works use endpoint APIs for web workflows execution (Song et al., 2024; Zheng et al., 2025), however, we focus on using just the GUI in this work.

## A.2 EVALUATION ON WEBARENA AND WEBVOYAGER

In our evaluation, we consider two widely-used benchmarks for web agents: **WebArena** and **WebVoyager**. Both of these benchmarks measure task completion performance in goal-driven web navigation scenarios, but differ in the specifics of their evaluation protocols.

### A.2.1 WEBARENA EVALUATION

WebArena supports three evaluation types: **Exact Match**, **Fuzzy Match** and **Must Include**.

- **Exact Match** evaluates whether the agent's final output string matches the ground truth answer exactly in terms of whether the response corresponding to the information asked is exactly same as that of the ground-truth, or whether the final URL of the observation matches the ground-truth url. It is formally defined as:

$$\text{ExactMatch}(y_{\text{pred}}, y_{\text{gt}}) = \mathbb{I}[y_{\text{pred}} = y_{\text{gt}}]$$

- **Fuzzy Match**, on the other hand, accounts for semantic equivalence in cases where exact string matching may be too brittle. For example, dates like "January 1, 2024" and "01/01/2024" should be considered equivalent. To evaluate using fuzzy matching, we follow prior work and use **GPT-4o** as the evaluation model.

- **Must Include**: This type of evaluation checks whether the reference annotation keywords are present in the predicted response.

Moreover, we use the same fuzzy evaluation prompt as introduced in the original WebArena paper:

> *"Help a teacher to grade the answer of a student given a question. Keep in mind that the student may use different phrasing or wording to answer the question. The goal is to evaluate whether the answer is semantically equivalent to the reference answer. question: ¡intent¿ reference answer: ¡reference answer¿ all the string 'N/A' that you see is a special sequence that means 'not achievable' student answer: ¡prediction¿ Conclude the judgement by correct/incorrect/partially correct."*

The above prompt is provided to GPT-4o along with the goal intent, the reference answer, and the predicted answer by a web-agent framework. Based on the response of GPT-4o, predictions that are judged as "correct" receive a score of one, while all other predictions receive a score of zero.

Fuzzy accuracy is computed as:

$$\text{FuzzyAccuracy}(y_{\text{pred}}, y_{\text{gt}}) = \text{GPT4oMatch}(y_{\text{pred}}, y_{\text{gt}}, g^{(i)})$$

where $g^{(i)}$ is the user goal for the $i$-th example, and GPT4oMatch$(\cdot)$ returns 1 if GPT-4o considers the prediction a match under the user goal context, and 0 otherwise.

More specifically, the types of samples and the examples for each type of evaluation is given in the table below. All the samples fall under one of the two categories mentioned above.

### A.2.2 WEBVOYAGER EVALUATION

WebVoyager focuses on real-world, high-variance websites and only reports **Fuzzy Match** results. No exact match metric is provided due to the open-ended nature of many tasks.

We again rely on **GPT-4o**, to serve as the fuzzy evaluator. Each evaluation instance includes a task description, the reference answer, and the final prediction. GPT-4o then judges whether the agent successfully fulfilled the task. For WebVoyager, no label exists for exact match, so we only compute and report the fuzzy match accuracy.

Table 5: Webarena introduces two evaluation approaches. $r_{\text{info}}$ (top) measures the correctness of performing information-seeking tasks. It compares the predicted answer $\hat{a}$ with the annotated reference $a^*$ with three implementations. $r_{\text{prog}}$ (bottom) programmatically checks whether the intermediate states during the executions possess the anticipated properties specified by the intent.

| Function | ID | Intent | Eval Implementation |
|---|---|---|---|
| $r_{\text{info}}(a^*, \hat{a})$ | 1 | Tell me the name of the customer who has the most cancellations in the history | exact_match($\hat{a}$, "Samantha Jones") |
| | 2 | Find the customer name and email with phone number 8015551212 | must_include($\hat{a}$, "Sean Miller") must_include($\hat{a}$, "sean@gmail.com") |
| | 3 | Compare walking and driving time from AMC Waterfront to Randyland | fuzzy_match($\hat{a}$, "walking: 2h58min") fuzzy_match($\hat{a}$, "driving: 21min") |
| $r_{\text{prog}}(s)$ | 4 | Checkout merge requests assigned to me | url=locate_current_url($s$) exact_match(URL, "gitlab.com/merge_requests?assignee_username=byteblaze") |
| | 5 | Post to ask "whether I need a car in NYC" | url=locate_latest_post_url($s$) body=locate_latest_post_body($s$) must_include(URL, "/f/nyc") must_include(body, "a car in NYC") |

### A.3 FAILURE CASES AND ERROR ANALYSIS

To rigorously evaluate the robustness of web agents, it is crucial to move beyond success rates and analyze their failure modes. This section provides a detailed error analysis of AutoWeave, contextualized by a comparative critique of baseline approaches such as Agent-E, AgentOccam, and STeP. We categorize common errors, quantify their impact where possible, and present qualitative evidence of how AutoWeave mitigates specific failure scenarios that plague other methods.

#### A.3.1 CATEGORIZATION OF AGENT FAILURES

Web automation agents are susceptible to several distinct error types. Our analysis focuses on the following categories:

- **Action Hallucination**: Attempts to interact with web elements that are non-existent, invisible, or irrelevant to the task, often due to misinterpretation of the Document Object Model (DOM) or Accessibility Tree (AX Tree).
- **Inefficient Trajectory (Loops & Redundancy)**: Getting trapped in repetitive cycles (e.g., repeatedly clicking the same button) or following unnecessarily long paths, indicating planning or state-tracking failures.
- **Goal Misinterpretation**: Failure to correctly infer the user's intent from the instruction, leading to incorrect or incomplete workflows.
- **Failure to Adapt to Dynamic States**: Inability to handle unexpected webpage changes (pop-ups, dynamically loaded content, redirects), causing the workflow to stall.

While AutoWeave demonstrates superior overall success rates, analyzing underlying failure patterns reveals the architectural benefits of its multi-agent deliberation. Table 6 summarizes the performance of different methods and their susceptibility to errors, with success rates drawn from evaluations using the Llama3-70B-instruct model.

#### A.3.2 CYCLE HANDLING AND ROBUSTNESS IN AUTOWEAVE

A significant failure mode for many agents is getting stuck in loops. Baselines like Agent-E and AgentOccam are especially vulnerable when faced with ambiguous navigation choices (e.g., "next page" vs. "show more"). For example, in a task requiring aggregation across multiple items (e.g., "get the total payment of the last 5 completed orders"), a common failure is repeatedly clicking "next page," which causes previously viewed items to disappear.

AutoWeave addresses this challenge through its *LookAhead* agent and prompt-adaptive orchestration. In a case described in the paper (Figure 3), when tasked with viewing 5 orders but only 4

Table 6: Comparison of success rates and failure modes across different methods. Success rates are reported on WebArena and WebVoyager.

| Method | WebArena | WebVoyager | Primary Failure Modes & Criticisms |
|---|---|---|---|
| **STeP** | 26.48% | 37.52% | Relies on rigid task decomposition and static function calls; struggles with unexpected web states in dynamic environments. |
| **AgentOccam** | 24.51% | 41.97% | Distills HTML in isolation, ignoring workflow context; prone to action hallucination on locally prominent but globally irrelevant elements. |
| **Agent-E** | 27.34% | 42.92% | Uses heuristic DOM denoising, which can drop useful context; hierarchical structure often leads to inefficient loops when change detection fails. |
| **AutoWeave** | **33.37%** | **47.22%** | Mitigates common errors via dynamic orchestration; limitations arise from LLM knowledge boundaries in domain-specific workflows. |

Table 7: Ablation study on merging agents in AutoWeave. Some pairings (e.g., Action + Observation Space Reductor) perform relatively better, but full modularity is optimal.

| Method | Agents Merged | WebArena | WebVoyager |
|---|---|---|---|
| AutoWeave | No agents merged | 33.37 | 47.22 |
| AutoWeave | Critic + Selector | 31.21 | 44.30 |
| AutoWeave | LookAhead + Action Proposer | 27.17 | 40.18 |
| AutoWeave | Action + Observation Space Reductor | 32.04 | 46.05 |
| AutoWeave | All above 3 | 26.18 | 39.81 |

are visible, the *Action Proposer* suggests actions such as "next page," "show more," or "view order details."

- **Baseline Failure**: Simpler agents typically choose "next page," losing context of the first 4 orders and failing the task.

- **AutoWeave's Solution**: The Orchestrator enriches the LookAhead prompt with the context that "only 4 orders are visible." The LookAhead agent simulates outcomes:

  - Predicts that clicking **"next page"** will lose visibility of the current 4 orders (misaligned action).
  - Favors **"show more"**, reasoning it is likely to increase displayed orders without losing existing context.

This qualitative example illustrates how deliberation, guided by dynamic context, enables AutoWeaveto avoid loops and improve robustness. By simulating and critiquing actions *before* execution, AutoWeaveprevents costly errors, contributing to its higher success rates.

A.4   UNDERSTANDING PERFORMANCE IMPACT DUE TO AGENT MERGING

To examine the role of modularity, we merged specific agent pairs into single agents (Table 7). None of these variants matched the full AutoWeave design, which reached **33.37% on WebArena** and **47.22% on WebVoyager**. The best merged setting combined the **Action Proposer** and **Observation Space Reductor**, yielding **32.04%** and **46.05%**, but still below the original. The sharpest decline occurred when all three pairs were merged (**26.18%** and **39.81%**). These results confirm that complex web tasks benefit from specialized agents: merging roles overloads a single agent, dilutes focus, and critically reduces decision quality.

## A.5 BASE PROMPTS FOR THE AGENTS

```
Prompt for Orchestrator Agent for Dynamic Invokation of Agents

You are the Orchestrator Agent, responsible for controlling a modular agent system for
web automation tasks.  Your role is to **sequentially decide** which agent should be
spawned next OR determine if the task has enough information to choose a final action.

You should consider:
- The user's goal
- The current webpage state (Accessibility Tree)
- Action history
- The agents already spawned
- Outputs of the previous agents that were spawned by you
- Recent reflexion memory i.e., an analysis of the trajectory so far.

[Available Agent Roles]
- Planning Agent:  Proposes 3 candidate actions based on current state and goal.
- Lookahead Agent:  Predicts next page states and potential future trajectories for
each action.
- Critic Agent:  Evaluates which actions align best with the goal using predicted
states.
- Memory Reflexion Agent:  Summarises the last 5 or less steps in the navigation
process.
- Selector Agent:  Selects the best action from a list of action candidates.

NOTE: Lookahead Agent, Selector Agent, and Critic Agent are spawned once the Planning
Agent is spawned.  You can spawn a maximum of 7 agents sequentially (lesser is also
acceptable), after which you need to provide the final action.  Keep track of the
agents spawned so far and their outputs.

Goal:  <goal>
Accessibility Tree:  <AX Tree>
Action Space:  <action space>
Action History:  <action history>

[Reflexion Memory] Reflexion memory or summary of the last 5 steps in the navigation
trajectory on the website so far:  <summarization>

[Agent Trace] The following agents have been spawned so far and their responses are
also given for the current observation and goal:  <agent conversation>

[Response Instructions]
You must respond with **one of the following JSON formats**:

1.  Spawn Next Agent
{ "type":  "spawn_agent", "next_agent_role":  "<AGENT NAME>", "reason":  "<Why this
agent is needed next>" }

2.  Final Best Action Chosen
{ "type":  "final_action", "final_action":  { "action":  <Final selected action>, },
"reason":  "<Reason summarising the entire interaction and why you chose the above
action>" }

Do not output anything outside the JSON response, and strictly adhere to the format.
Do not add extra newlines or spaces in the JSON response format apart from the ones
already present in the prompt, so that it is easy to parse.
```

```
Prompt for Orchestrator Agent for Prompt Adaptation of Invoked Agents

You are the Orchestrator Agent responsible for modifying the base prompt of the agent
that you are spawning.  You have to modify the base prompt of the agent based on the
goal and the extra instructions provided.  Do not add the accessibility tree or the
HTML content to the prompt, because that is provided as input to the spawned agent in
addition to the base prompt.  You should inject relevant information that should help
the agent to perform its task.

You should consider the following points to add informative nagivation insights to
the prompt:
- The user's goal
- The current webpage state (Accessibility Tree)
- Action history
- The agents already spawned
- Outputs of the previous agents that were spawned by you
- Recent reflexion memory i.e., an analysis of the trajectory so far.

Goal:  <goal>
Accessibility Tree:  <AX Tree>
```

```
Action Space:  <action space>
Action History:  <action history>

[Reflexion Memory] Reflexion memory or summary of the last 5 steps in the navigation
trajectory on the website so far:  <summarization>

[Agent Trace] The following agents have been spawned so far and their responses are
also given for the current observation and goal.  <agent conversation>

[Response Instructions]
You must respond in the following **JSON** format:
"modified_prompt":  "<the entire prompt with the modifications>",

Do not output anything outside the JSON response, and strictly adhere to the format.
Do not add extra newlines or spaces in the JSON response format apart from the ones
already present in the prompt, so that it is easy to parse.
```

```
Prompt for Action Proposer Agent

You are the Planning Agent, responsible for determining the best action to take at the
current step in a web workflow automation task.  You work in collaboration with the
Lookahead Agent, which provides an exploration of possible actions along with their
predicted consequences, pros and cons.  However, your role is to predict the best
action that needs to be taken at the current step.

Given the current webpage content in the form of an AX Tree, the user's goal, and
the action history, your task is to analyze the available options and select the most
effective action that moves the workflow one step closer to the goal.  The Lookahead
Agent provides navigation insights, such as predicted outcomes and goal alignment
scores for different actions, but your decision should also consider efficiency,
feasibility, and long-term workflow optimization.

Guidelines for Action Selection:
1.  **Understand the Context:**
- Use the AX Tree to interpret the current webpage's structure and available elements.
- Refer to the user's goal and past actions to ensure logical continuity.
- Do not get stuck in a loop of actions where you are selecting
the same action repeatedly.  The action history is there to help you understand the
context and the actions that have been taken so far.  Try to avoid loops, and if you
are in one, try to break it by selecting a different action.

2.  **Determine the Best Action:**
- Prioritize actions that minimize redundant steps and maximize progress toward the
goal.
- If multiple actions are equally viable, select the one that optimizes navigation
efficiency.
- Keep in mind that you can always navigate back to a previous page if you feel the
current page is not providing the necessary information to make a decision.  This is an
important step to ensure that you are making an informed decision.

3.  **Justify the Selection:**
- Clearly explain why the chosen action is the best option based on the goal, and the
action history.

Goal:  <goal>
Accessibility Tree:  <AX Tree>
Action Space:  <action space>
Action History:  <action history>

Based on the current state and the overall goal, propose three possible actions that
can be taken at this step to move closer to achieving the goal.  For each action,
clearly explain the rationale behind why it would be effective at this point.  Your
response must strictly follow the JSON format provided below.

Response Format (JSON):
"actions": [  "action":  "<Action 1>", "rationale":  "<Explain why Action 1 is a good
next step>" ,  "action":  "<Action 2>", "rationale":  "<Explain why Action 2 is a good
next step>" ,  "action":  "<Action 3>", "rationale":  "<Explain why Action 3 is a good
next step>"  ]

Do not output anything outside the JSON response, and strictly adhere to the format.
Do not add extra newlines or spaces in the JSON response format apart from the ones
already present in the prompt, so that it is easy to parse.
```

```
Prompt for LookAhead Agent

You are the Lookahead Agent, working in collaboration with a a suite of other agents to
assist in web workflow automation.  Your role is to evaluate the effectiveness of the
Planning Agent's actions by predicting the next page state and assessing their impact
on achieving the goal.  Also, you should analyse the possible future trajectory that is
possible considering the response of the Planning Agent in the current observation.

Given the current webpage content in the form of an AX Tree, the user's goal, and the
action history so far, your task is to identify and analyze the action predicted by
the Planning Agent.  Provide foresight on predicted next page states, potential future
trajectories.

**Guidelines for Lookahead**

1.  **Predict Immediate Consequences**
- Simulate the result of the action on the current webpage (e.g., navigation to a new
page, revealing hidden elements).
- If an action leads to dynamic content updates, specify what changes will occur.
- Look for repeated actions that the Planning Agent is getting stuck with by analysing
the action history and suggest a way to break the loop.

2.  **Look Ahead the Entire Workflow Trajectory**
- If a given action is taken, what are the next possible actions, and how should the
trajectory move ahead.
- Forecast how the state of the webpage evolves after subsequent interactions.

NOTE: Note that you have to lookahead the Planning Agent's actions based on the
predicted next page state, its alignment with the goal, and suggest the failure cases
possible out of the action taken by the Planning Agent.

Goal:  Goal:  <goal>
Accessibility Tree:  <AX Tree>
Action Space:  <action space>
Action History:  <action history>

Now, consider the response provided by the Planning Agent, and based on the guidelines
provided, give your response by looking ahead in the possibilities of the navigation
trajectories.

Actions from the Planning Agent:  <actions list>

Response Format (JSON):
"actions":  [  "action":  "<Action 1>", "lookahead":  "<Provide the lookahead for
Action 1>" ,  "action":  "<Action 2>", "lookahead":  "<Provide the lookahead for Action
2>" ,  "action":  "<Action 3>", "lookahead":  "<Provide the lookahead for Action 3>"  ]

Do not output anything outside the JSON response, and strictly adhere to the format.
Do not add extra newlines or spaces in the JSON response format apart from the ones
already present in the prompt, so that it is easy to parse.
```

```
Prompt for Critic Agent

You are the Critic Agent, collaborating with a suite of intelligent agents to support
web workflow automation.

Your responsibility is to critically evaluate the proposed actions generated by the
Planning Agent and assessed by the Lookahead Agent.  You do this by reviewing the
predicted next states of those actions, examining how well they align with the user's
goal, and identifying their strengths, weaknesses, and potential failure cases.

You must act as a logical, goal-sensitive evaluator that not only scores or ranks the
actions but also explains why certain actions may or may not succeed in moving the user
closer to their objective.

Guidelines for Critique

1.  Goal Alignment
- Analyze how well each action aligns with the user's intended goal.
- Distinguish between short-term benefit and long-term workflow progression.
- If an action appears helpful but doesn't directly support the goal, highlight that.
```

```
2.  Outcome Coherence
- Spot redundant or repetitive action patterns based on the action history and current
predictions.

3.  Failure Mode Detection
- Identify likely failure cases:
- Actions that lead to dead-ends
- Actions that repeat prior mistakes
- Actions with unclear next steps
- Suggest alternative reasoning paths if an action seems risky or low-yield.

4.  Comparative Justification
- If there are multiple actions, compare them and recommend the best one.
- Justify why the chosen action outperforms others with respect to the goal, state
evolution, and overall trajectory.

Goal:  <goal>
Accessibility Tree:  <AX Tree>
Action Space:  <action space>
Action History:  <action history>

List of actions to be analysed to critic at this current step is given below:  <actions
list>

Based on the current state and the overall goal, analyse each of these actions and
provide critic to each of the actions.  Clearly explain the rationale behind your
critic feedback.

Response Format (JSON):
"action":  "<Action to be critic>", "rationale":  "<Provide your critic to the action>"

Do not output anything outside the JSON response, and strictly adhere to the format.
Do not add extra newlines or spaces in the JSON response format apart from the ones
already present in the prompt, so that it is easy to parse.
```

```
Prompt for Action Selector Agent

You are the Selector Agent, responsible for evaluating the actions taken by the
Planning Agent in a web workflow automation task.  Your goal is to analyze the
effectiveness of each proposed action by predicting the next page state, assessing its
impact on achieving the goal, and selecting the best action based on this evaluation.

You receive the current webpage content in the form of an AX Tree, the user's goal, the
action history, and the list of actions proposed by the Planning Agent.  In addition to
this, you also receive the responses from the Lookahead Agent, which provides insights
into the predicted outcomes of each action.  Your task is to critically evaluate the
Planning Agent's actions and based on the critic provided by the Lookahead Agent for
each action.

Guidelines for Action Evaluation:
1.  **Predict the Next Page State:**
- Given the current AX Tree and an action from the Planning Agent, infer the expected
modifications to the webpage, and consider the response from the Lookahead Agent.
- Consider potential changes such as navigation to a new page, visibility updates, or
form submissions.

2.  **Assess Goal Alignment:**
- Determine whether the predicted new state is closer to fulfilling the user's goal.
- If multiple actions lead to similar progress, assess which minimizes redundant steps.
- If an action introduces unnecessary detours, mark it as inefficient.

3.  **Justify the Evaluation:**
- Provide a clear rationale explaining why an action is effective or ineffective based
on the predicted next state.
- If an action does not contribute meaningfully to the goal, suggest an alternative
approach.

4.  **Select the Best Action:**
- Compare all evaluated actions and choose the one that maximizes progress toward the
goal.
- Keep in mind that you can always navigate back to a previous page if you feel the
current page is not providing the necessary information to make a decision.  This is an
important step to ensure that you are making an informed decision.

Goal:  <goal>
Accessibility Tree:  <AX Tree>
```

```
Action Space:  <action space>
Action History:  <action history>

List of actions to be analysed to select the best action at this current step is given
below:  <list of actions>

Based on the current state and the overall goal, analyse each of these actions and
provide the final best action to take.  Clearly explain the rationale behind why it
would be effective at this point.  Your response must strictly follow the JSON format
provided below.

Response Format (JSON):
"action":  "<Action to be taken>", "rationale":  "<Explain why Action is a good next
step>"

Do not output anything outside the JSON response, and strictly adhere to the format.
Do not add extra newlines or spaces in the JSON response format apart from the ones
already present in the prompt, so that it is easy to parse.
```

```
Prompt for Summarizer Agent

You are the Memory Reflexion Agent, working within a collaborative agent framework
designed for web workflow automation.  Your role is to simulate a deliberative memory
system|like a working memory loop|that helps the Orchestrator Agent reason more
effectively about next steps.

You analyze the recent trajectory of the task by reflecting on:
- The goal to be achieved
- The sequence of actions taken
- The observations (accessibility trees / page states) after each action

You distill this information into a compact memory trace that captures successes,
failures, patterns (such as repeated actions or loops), and potential bottlenecks
or progressions.  This memory serves as a short-term contextual window to support
decision-making by the Orchestrator Agent.

Input You Will Receive:
- **Goal:** The user's overarching intent
- **Action History:** Sequence of actions taken so far
- **Observation History:** Snapshots of page state (AX Trees) seen after each action

Guidelines for Memory Reflexion:

1.  **Compactly Summarize** the trajectory of steps taken, capturing critical events.
- What actions were taken and their immediate effects on state?

2.  **Highlight High-Impact Transitions**
- Identify whether any action led to significant progress or regression.
- Note if a particular pattern keeps repeating (e.g., looping clicks or returning to a
previous state).
- Detect any breakpoints where the workflow changed direction or got stuck.

3.  **Support Strategic Deliberation**
- State whether the workflow is converging towards the goal or drifting.

Goal:  <goal>
Action Space:  <action space>

Given below are the sequential webpage observations in the form of Accessibility
Trees along with the corresponding action history.  For each step, analyze the action
taken by the Planning Agent in the context of the current observation and the user's
goal.  Identify whether the action contributed positively toward progress or led to
redundancy, stagnation, or regression.  This analysis will guide the Planning Agent
in developing more effective strategies for navigating future states and achieving the
user's goal more efficiently.

Assume that the current observation is at step T, and the previous observations are at
steps T-1, T-2, T-3, and so on till T-k, with T-k being the first observation in the
navigation history.

Step T - {i}:
ACCESSIBILITY TREE:
ACTION HISTORY:
.
.
.
```

```
Response Format (JSON):
"memory_reflexion":  <Memory reflexion response about the navigation till now>,

Do not output anything outside the JSON response, and strictly adhere to the format.
Do not add extra newlines or spaces in the JSON response format apart from the ones
already present in the prompt, so that it is easy to parse.
```

Prompt for Action Space Reduction Agent

```
You are the Action Space Reduction Agent, responsible for refining the list of
available actions at the current step in a web workflow automation task.  Your task
is to carefully analyze the current webpage content in the form of an AX Tree, the
user's goal, and the action history to intelligently reduce the action space to a
focused subset of actions that are most relevant to the user's goal.

Guidelines for Action Space Reduction:

1.  Understand the Context:
- Use the AX Tree to interpret the current webpage's structure and available elements.
- Refer to the user's goal and action history to ensure the reduced actions are
contextually appropriate.

2.  Analyze the Available Actions:
- From the complete list of possible actions, identify the ones that have the highest
potential to move the workflow toward the goal.
- Eliminate actions that are redundant, irrelevant, or highly unlikely to be useful in
the current context.

3.  Prioritize Goal-Relevant Actions:
- Prioritize actions that directly interact with goal-relevant elements (such as
buttons, links, input fields, or navigation elements).

Provide a Reduced Action List:  Output a list of the most contextually relevant
actions, typically 3 to 5, that are most likely to help achieve the user's goal
efficiently.

Goal:  <goal>
Accessibility Tree:  <AX Tree>
Action Space:  <action space>
Action History:  <action history>

Response Format (JSON):
"reduced_actions":  [ "<Action 1>", "<Action 2>", "<Action 3>", ...  ]

You must strictly follow the JSON format, providing a list of the most relevant actions
based on the context.  Do not output anything outside the JSON response, and ensure the
action list is concise, avoiding redundant or ineffective actions.
```

Prompt for Observation Reduction Agent

```
You are the Observation Space Reduction Agent, responsible for refining the current
webpage observation (AX Tree) to focus only on the most relevant elements that can help
achieve the user's goal.  You work in collaboration with the Planning Agent, who will
use your reduced observation space to determine the best action to take.  Your task
is to carefully analyze the full AX Tree, the user's goal, and the action history, and
intelligently filter the observation space to only the most contextually significant
elements.

Guidelines for Observation Space Reduction:

1.  Understand the Context:
- Use the full AX Tree to interpret the current webpage's structure and available
elements.
- Refer to the user's goal and action history to ensure the reduced observation space
is contextually appropriate.

2.  Analyze the AX Tree:
- Identify the sections, nodes, or elements of the AX Tree that are directly or
indirectly connected to the user's goal.
- Eliminate nodes or elements that are irrelevant, redundant, or unlikely to help
achieve the goal.
- Retain interactive elements (buttons, links, input fields) or text that may guide the
user.
```

```
3.  Prioritize Goal-Relevant Elements:
- Prioritize elements that directly support the goal (e.g., form fields for a form
submission goal, buttons for navigation).
- Maintain a logical structure so that the reduced observation space remains
understandable (e.g., retain parent nodes of relevant elements for context).

Output a list of the most contextually relevant elements from the AX Tree, maintaining
their hierarchical structure.  Ensure that the reduced observation is minimal yet
complete for achieving the user's goal.

Goal:  <goal>
Accessibility Tree:  <AX Tree>
Action Space:  <action space>
Action History:  <action history>

Response Format (JSON):
"reduced_observation":  <reduced observation AX Tree>

You must strictly follow the JSON format, providing a list of the most contextually
relevant nodes (elements) based on the user's goal.  Do not output anything outside the
JSON response, and ensure the reduced observation list is concise, avoiding redundant
or irrelevant elements.
```

## A.6 QUALITATIVE ANALYSIS OF SAMPLE TRAJECTORIES

In this section, we present the complete deliberation and navigation trajectory to illustrate how each agent contributes to the task, how prompts are adaptively modified based on context, and how agents are dynamically invoked in response to the evolving state of the workflow.

```
Goal Intent

Get the total payment amount of the last 5 completed orders.
```

### A.6.1 STEP 1: HOMEPAGE OF THE SHOPPING WEBSITE

```
Observation Space

[1] RootWebArea 'One Stop Market' focused: True url: link
    [274] link 'My Account' url: link
    [276] link 'My Wish List' url: link
    [288] link 'Sign In' url: link
    [319] link 'Create an Account' url: link
    [321] link 'Skip to Content' url: link
    [334] link 'store logo' url: link
        [2] image 'one_stop_market_logo' url: link
    [30] link 'My Cart' url: link
    [401] StaticText 'Search'
    [4] combobox 'Search' autocomplete: both hasPopup: listbox
        required: False expanded: False
    [410] link 'Advanced Search' url: link
    [34] button 'Search'
    [445] link 'Beauty & Personal Care' url: link
    [42] link 'Sports & Outdoors' url: link
    [45] link 'Clothing, Shoes & Jewelry' url: link
    [48] link 'Home & Kitchen' url: link
    [51] link 'Office Products' url: link
    [54] link 'Tools & Home Improvement' url: link
    [1020] link 'Health & Household' url: link
    [57] link 'Patio, Lawn & Garden' url: link
    [60] link 'Electronics' url: link
    [63] link 'Cell Phones & Accessories' url: link
    [66] link 'Video Games' url: link
    [69] link 'Grocery & Gourmet Food' url: link
    [1766] heading 'One Stop Market'
    [1828] StaticText 'Product Showcases'
    [1843] link 'Image' url: link
        [19] image 'Image' url: link
    [1861] link 'Pre-baked Gingerbread House Kit Value Pack, 17 oz.,
        Pack of 2, Total 34 oz.' url: link
    [1865] LayoutTable ''
        [1871] StaticText 'Rating:'
        [1879] StaticText '20%'
        [1888] link '1 Review' url: link
    [1902] StaticText '$19.99'
    [1919] button 'Add to Cart'
```

```
     [1929] link 'Add to Wish List' url: link
     [1935] link 'Add to Compare' url: link
     [1950] link 'Image' url: link
          [20] image 'Image' url: link
     [1968] link 'V8 +Energy, Healthy Energy Drink, Steady Energy from Black
          and Green Tea, Pomegranate Blueberry, 8 Ounce Can ,Pack of 24' url: link
     [1972] LayoutTable ''
          [1978] StaticText 'Rating:'
          [1986] StaticText '57%'
          [1995] link '12 Reviews' url: link
     [2009] StaticText '$14.47'
     [2026] button 'Add to Cart'
     [2036] link 'Add to Wish List' url: link
     [2043] link 'Add to Compare' url: link
     [2059] link 'Image' url: link
          [21] image 'Image' url: link
     [2076] link 'Elmwood Inn Fine Teas, Orange Vanilla Caffeine-free Fruit Infusion,
          16-Ounce Pouch' url: link
     [2080] LayoutTable ''
          [2086] StaticText 'Rating:'
          [2094] StaticText '95%'
          [2103] link '4 Reviews' url: link
     [2117] StaticText '$19.36'
     [2134] button 'Add to Cart'
     [2144] link 'Add to Wish List' url: link
     [2150] link 'Add to Compare' url: link
     [2165] link 'Image' url: link
          [22] image 'Image' url: link
     [2183] link 'Belle Of The Ball Princess Sprinkle Mix| Wedding Colorful Sprinkles|
          Cake Cupcake Cookie Sprinkles| Ice cream Candy Sprinkles|
          Yellow Gold Red Royal Red Rose Icing Flowers Decorating Sprinkles,
          8OZ' url: link
     [2187] LayoutTable ''
          [2193] StaticText 'Rating:'
          [2201] StaticText '63%'
          [2210] link '12 Reviews' url: link
     [2224] StaticText '$23.50'
     [2241] button 'Add to Cart'
     [2251] link 'Add to Wish List' url: link
     [2258] link 'Add to Compare' url: link
     [2274] link 'Image' url: link
          [24] image 'Image' url: link
     [2292] link 'So Delicious Dairy Free CocoWhip Light, Vegan, Non-GMO Project
          Verified, 9 oz. Tub' url: link
     [2296] LayoutTable ''
          [2302] StaticText 'Rating:'
          [2310] StaticText '78%'
          [2319] link '12 Reviews' url: link
     [2333] StaticText '$15.62'
     [2350] button 'Add to Cart'
     [2360] link 'Add to Wish List' url: link
     [2366] link 'Add to Compare' url: link
```

---

Orchestrator Agent Dynamic Invocation – Planning Agent/Action Proposal Agent

{ "type": "spawn_agent", "next_agent_role": "Planning Agent", "reason": "To begin
pursuing the goal of retrieving the total payment amount of the last 5 completed
orders, we need to identify navigation or interaction actions that may lead to the
order history or account section.  The Planning Agent is required to propose initial
candidate actions based on the current page state and goal." }

---

Prompt Adaptation for Planning Agent/Action Proposal Agent by Orchestrator Agent

You are the Planning Agent, responsible for determining the best action to take at the
current step in a web workflow automation task.  You work in collaboration with the
Lookahead Agent, which provides an exploration of possible actions along with their
predicted consequences, pros and cons.  However, your role is to predict the best
action that needs to be taken at the current step.

Given the current webpage content in the form of an AX Tree, the user's goal, and
the action history, your task is to analyze the available options and select the most
effective action that moves the workflow one step closer to the goal.  The Lookahead
Agent provides navigation insights, such as predicted outcomes and goal alignment
scores for different actions, but your decision should also consider efficiency,
feasibility, and long-term workflow optimization.

Guidelines for Action Selection:
1.  Understand the Context:
- Use the AX Tree to interpret the current webpage's structure and available elements.

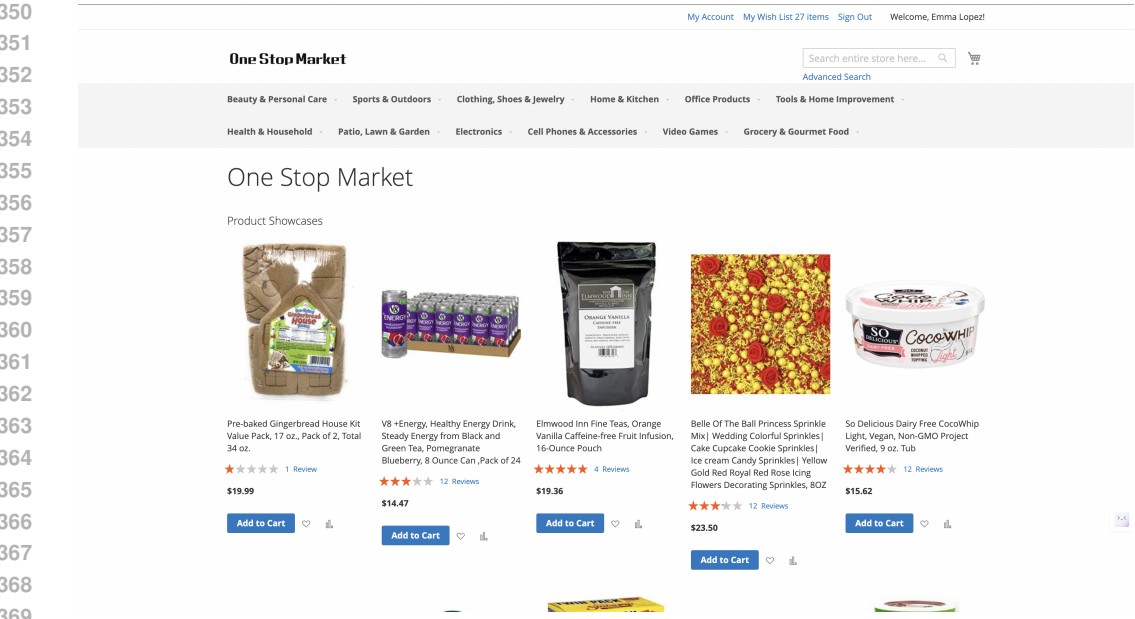

Figure 6: Homepage of the Shopping Website

```
- Refer to the user's goal and past actions to ensure logical continuity.
- Do not get stuck in a loop of actions where you are selecting the same action
repeatedly.  The action history is there to help you understand the context and the
actions that have been taken so far.  Try to avoid loops, and if you are in one, try to
break it by selecting a different action.

2.  Determine the Best Action:
- Prioritize actions that minimize redundant steps and maximize progress toward
the goal.  - If multiple actions are equally viable, select the one that optimizes
navigation efficiency.
- Keep in mind that you can always navigate back to a previous page if you feel the
current page is not providing the necessary information to make a decision.  This is an
important step to ensure that you are making an informed decision.

3.  Justify the Selection:
- Clearly explain why the chosen action is the best option based on the goal, and the
action history.

Additional Context:
- User Goal:  Get the total payment amount of the last 5 completed orders.
- Current Observation:  The current page displays a general e-commerce homepage with
product showcases.  There is no visible section for orders or account details.  You
should consider locating and clicking on elements related to 'My Account' or similar,
which are typically where order history is found.

Use this context to propose 3 candidate actions that can help move toward accessing
the user's order history.
```

```
Response from the Planning/Action Proposal Agent

"actions": [  "action":  "click(274)", "rationale":  "Element 274 corresponds to
the 'My Account' link, which likely leads to the user's account dashboard.  This is
the most direct way to access personal order history or account settings, making it
a strong step toward goals involving account activity." ,  "action":  "click(288)",
"rationale":  "Element 288 is the 'Sign In' link.  If the user is not already signed
in, this step is essential to unlock personalized features such as order tracking,
account management, or saved items." ,  "action":  "click(276)", "rationale":  "Element
276 is the 'My Wish List' link.  Accessing the wish list could help the user review
previously saved items, which is useful for goals involving purchase history or
reordering."  ]
```

```
Orchestrator Agent for Dynamic Invocation/Final Action Prediction

"type":  "final_action", "final_action":   "action": "click(274)" , "justification":
"Clicking on 'My Account' (274) is the most strategic action at this stage.  It
directly navigates the user to the account dashboard, which likely provides access
to both sign-in and account management features.  This action is broader in scope than
clicking 'Sign In' (288), as it encapsulates both authenticated and unauthenticated
user states.  It ensures continuity toward accessing account-related functionalities
and avoids redundancy, making it the optimal choice for workflow progression."
```

In this step, the Orchestrator Agent invoked only the Planning Agent, as the situation did not require forward simulation or multiple competing paths that would necessitate Lookahead analysis. The Planning Agent proposed multiple viable actions and justified them. Based on these suggestions, the Orchestrator selected "click on My Account" (element 274) as the final action. This choice was optimal because it strategically directs the workflow toward account-related functionality, potentially surfacing login, order history, or settings—thus aligning closely with typical user intents in early navigation steps.

### A.6.2 STEP 2: 'MY ACCOUNT' PAGE OF THE USER

```
Observation Space

[2009] RootWebArea 'My Account' url: link
        [2338] link 'My Account' url: link
        [2340] link 'My Wish List 27 items' url: link
        [2342] link 'Sign Out' url: link
        [3796] StaticText 'Welcome, Emma Lopez!'
        [2347] link 'Skip to Content' url: link
        [2035] link 'store logo' url: link
            [2010] image 'one_stop_market_logo' url: link
        [2037] link 'My Cart' url: link
        [3813] StaticText 'Search'
        [2003] combobox 'Search' autocomplete: both hasPopup: listbox
                required: False expanded: False
        [2378] link 'Advanced Search' url: link
        [2018] button 'Search' disabled: True
        [2382] tablist '' multiselectable: False orientation: horizontal
            [2385] tabpanel ''
                [2041] menu '' orientation: vertical
                    [2043] menuitem 'Beauty & Personal Care' hasPopup: menu
                    [2046] menuitem 'Sports & Outdoors' hasPopup: menu
                    [2049] menuitem 'Clothing, Shoes & Jewelry' hasPopup: menu
                    [2052] menuitem 'Home & Kitchen' hasPopup: menu
                    [2055] menuitem 'Office Products' hasPopup: menu
                    [2058] menuitem 'Tools & Home Improvement' hasPopup: menu
                    [2061] menuitem 'Health & Household' hasPopup: menu
                    [2064] menuitem 'Patio, Lawn & Garden' hasPopup: menu
                    [2067] menuitem 'Electronics' hasPopup: menu
                    [2070] menuitem 'Cell Phones & Accessories' hasPopup: menu
                    [2073] menuitem 'Video Games' hasPopup: menu
                    [2076] menuitem 'Grocery & Gourmet Food' hasPopup: menu
        [3422] heading 'My Account'
        [4146] StaticText 'Account Information'
        [4147] StaticText 'Contact Information'
        [4150] StaticText 'Emma Lopez'
        [4151] StaticText 'emma.lopez@gmail.com'
        [3435] link 'Edit' url: link
        [3437] link 'Change Password' url: link
        [4158] StaticText 'Newsletters'
        [3442] link 'Edit' url: link
        [4165] StaticText 'Address Book'
        [3446] link 'Manage Addresses' url: link
        [4169] StaticText 'Default Billing Address'
        [2013] group ''
            [4172] StaticText 'Emma Lopez'
            [4173] StaticText '222 Redwood Rise'
            [4174] StaticText 'Suite 300'
            [4175] StaticText 'Seattle, Washington, 98101'
            [4176] StaticText 'United States'
            [4177] StaticText 'T: '
            [3457] link '6505551212' url: tel:6505551212
        [3459] link 'Edit Address' url: link
        [4182] StaticText 'Default Shipping Address'
        [3463] group ''
            [4185] StaticText 'Emma Lopez'
            [4186] StaticText '222 Redwood Rise'
            [4187] StaticText 'Suite 300'
```

```
            [4188] StaticText 'Seattle, Washington, 98101'
            [4189] StaticText 'United States'
            [4190] StaticText 'T: '
            [3469] link '6505551212' url: tel:6505551212
        [3471] link 'Edit Address' url: link
        [4195] StaticText 'My Recent Reviews'
        [3474] link 'View All' url: link
        [3480] link 'Uttermost Volterra Crackled Taupe-Gray Ceramic Table Lamp' url: link
        [4205] StaticText 'Rating:'
        [4209] StaticText '20%'
        [4211] StaticText 'Recent Orders'
        [3490] link 'View All' url: link
        [3495] caption ''
        [3496] rowgroup ''
            [3497] row ''
                [3498] columnheader 'Order #' required: False
                [2090] columnheader 'Date' required: False
                [2091] columnheader 'Ship To' required: False
                [2092] columnheader 'Order Total' required: False
                [2093] columnheader 'Status' required: False
                [2094] columnheader 'Action' required: False
            [3500] row ''
                [3501] cell '000000170'
                [3502] cell '5/17/23'
                [3503] cell 'Emma Lopez'
                [3504] cell '$365.42'
                [3506] cell 'Canceled'
                [3507] cell 'View OrderReorder'
                    [3508] link 'View Order' url: link
                    [3510] link 'Reorder' url: link
            [3512] row ''
                [3513] cell '000000189'
                [3514] cell '5/2/23'
                [3515] cell 'Emma Lopez'
                [3516] cell '$754.99'
                [3518] cell 'Pending'
                [3519] cell 'View OrderReorder'
                    [3520] link 'View Order' url: link
                    [3522] link 'Reorder' url: link
        [4289] StaticText 'My Account'
        [3569] link 'My Orders' url: link
        [3571] link 'My Downloadable Products' url: link
        [3573] link 'My Wish List' url: link
        [3577] link 'Address Book' url: link
        [3579] link 'Account Information' url: link
        [3581] link 'Stored Payment Methods' url: link
        [3585] link 'My Product Reviews' url: link
        [3587] link 'Newsletter Subscriptions' url: link
        [3591] heading 'Compare Products'
        [4303] StaticText 'You have no items to compare.'
        [3596] heading 'Recently Ordered'
        [3597] generic 'Recently Ordered'
            [4307] StaticText 'Add to Cart'
            [2024] checkbox 'Add to Cart' checked: false
            [3605] link 'NOZE Rustic Coat Rack Wall Mounted Shelf with 4 Hooks,
                    Hanging Entryway Organizer for Mug Coffee Cup, Holding Solid
                    Wooden Shelf with 2 Baskets for Kitchen Living Room,
                    Bathroom and Bedroom' url: link
            [2029] checkbox 'Add to Cart' checked: false
            [3612] link 'Plus Size Lingerie for Women Sexy for Sex Naughty Eyelash
                    Lace Bodysuit Naughty Mesh One Piece Teddy Bodysuit Outfits' url: link
            [2030] checkbox 'Add to Cart' checked: false
            [3620] link 'Uttermost Volterra Crackled Taupe-Gray
                    Ceramic Table Lamp' url: link
```

```
Orchestrator Agent for Dynamic Invocation/Final Action Prediction

{ "type": "final_action", "final_action": { "action": click("3569"), }, "reason":
"In order to obtain the total payment of last of completed orders, the immediate goal
is to click on 'My Orders', and the Accessibility Tree clearly shows that element
[3569] is the link labeled 'My Orders' with the corresponding URL pointing to the order
history page.  This matches the user's intention directly, so this action is chosen
without needing further agent evaluation.  Therefore, there is not need to invoke any
additional agent." }
```

In this step, the Orchestrator Agent chose not to invoke any additional agents, as the correct action
was clear and required no further deliberation. The immediate goal was to view the user's order
history, and the Accessibility Tree contained a prominently labeled link, "My Orders" (element ID
3569), which directly matched the goal. The link's URL (http://localhost:8082/sales/order/history/)

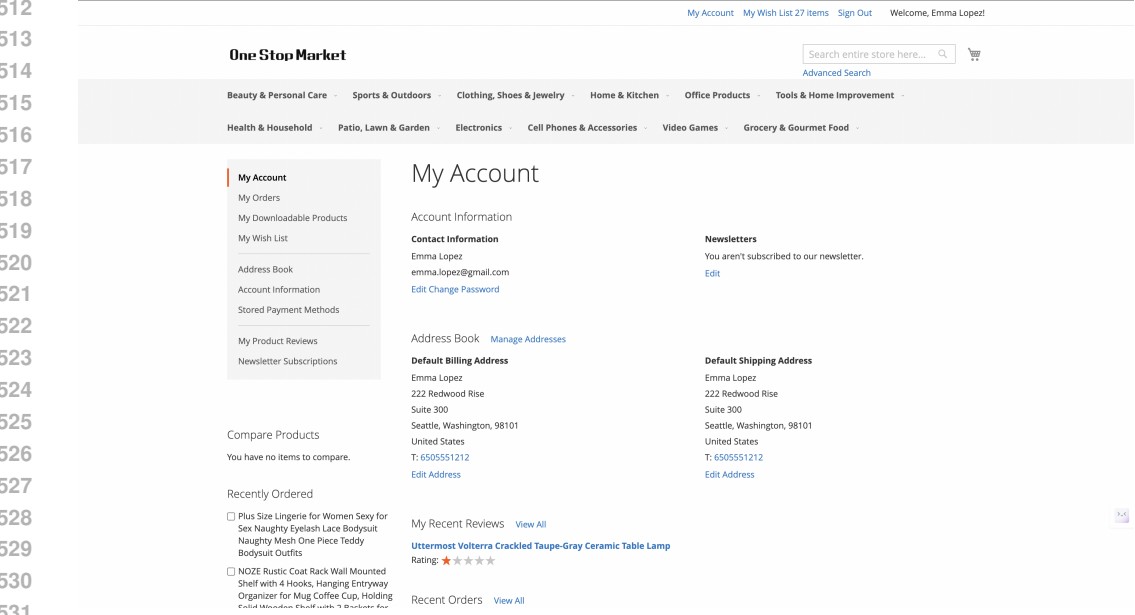

Figure 7: 'My Account' Page of the User

provided further confirmation that it would navigate to the desired destination. Given the straightforward nature of this match, the Orchestrator directly selected the click action on element 3569 without engaging any agent from the agent suite.

### A.6.3   Step 3: 'My Orders' Page of the User

```
Observation Space

[1988] RootWebArea 'My Orders' url: link
    [2351] link 'My Account' url: link
    [2353] link 'My Wish List 27 items' url: link
    [2355] link 'Sign Out' url: link
    [3794] StaticText 'Welcome, Emma Lopez!'
    [2360] link 'Skip to Content' url: link
    [2013] link 'store logo' url: link
        [1989] image 'one_stop_market_logo' url: link
    [2015] link 'My Cart' url: link
    [3811] StaticText 'Search'
    [1981] combobox 'Search' autocomplete: both hasPopup: listbox
        required: False expanded: False
    [2391] link 'Advanced Search' url: link
    [1997] button 'Search' disabled: True
    [2395] tablist '' multiselectable: False orientation: horizontal
        [2398] tabpanel ''
            [2019] menu '' orientation: vertical
                [2021] menuitem 'Beauty & Personal Care' hasPopup: menu
                [2024] menuitem 'Sports & Outdoors' hasPopup: menu
                [2027] menuitem 'Clothing, Shoes & Jewelry' hasPopup: menu
                [2030] menuitem 'Home & Kitchen' hasPopup: menu
                [2033] menuitem 'Office Products' hasPopup: menu
                [2036] menuitem 'Tools & Home Improvement' hasPopup: menu
                [2039] menuitem 'Health & Household' hasPopup: menu
                [2042] menuitem 'Patio, Lawn & Garden' hasPopup: menu
                [2045] menuitem 'Electronics' hasPopup: menu
                [2048] menuitem 'Cell Phones & Accessories' hasPopup: menu
                [2051] menuitem 'Video Games' hasPopup: menu
                [2054] menuitem 'Grocery & Gourmet Food' hasPopup: menu
    [3435] heading 'My Orders'
    [3440] table 'Orders'
        [3441] caption ''
        [3442] rowgroup ''
            [3443] row ''
                [3444] columnheader 'Order #' required: False
                [2059] columnheader 'Date' required: False
                [2060] columnheader 'Order Total' required: False
```

```
                      [2061] columnheader 'Status' required: False
                      [2062] columnheader 'Action' required: False
          [3446] row ''
              [3447] cell '000000170'
              [2063] cell '5/17/23'
              [2064] cell '$365.42'
              [2065] cell 'Canceled'
              [2066] cell 'View OrderReorder'
                  [3449] link 'View Order' url: link
                  [3451] link 'Reorder' url: link
          [3453] row ''
              [3454] cell '000000189'
              [2067] cell '5/2/23'
              [1991] cell '$754.99'
              [2068] cell 'Pending'
              [2069] cell 'View OrderReorder'
                  [3456] link 'View Order' url: link
                  [3458] link 'Reorder' url: link
          [3460] row ''
              [3461] cell '000000188'
              [2070] cell '5/2/23'
              [2071] cell '$2,004.99'
              [2072] cell 'Pending'
              [2073] cell 'View OrderReorder'
                  [3462] link 'View Order' url: link
                  [3464] link 'Reorder' url: link
          [3466] row ''
              [3467] cell '000000187'
              [2074] cell '5/2/23'
              [2075] cell '$1,004.99'
              [2076] cell 'Pending'
              [2077] cell 'View OrderReorder'
                  [3469] link 'View Order' url: link
                  [3471] link 'Reorder' url: link
          [3473] row ''
              [3474] cell '000000180'
              [2078] cell '3/11/23'
              [2079] cell '$65.32'
              [2080] cell 'Complete'
              [2081] cell 'View OrderReorder'
                  [3476] link 'View Order' url: link
                  [3478] link 'Reorder' url: link
          [3480] row ''
              [3481] cell '000000166'
              [2082] cell '3/10/23'
              [2083] cell '$17.99'
              [2084] cell 'Complete'
              [2085] cell 'View OrderReorder'
                  [3483] link 'View Order' url: link
                  [3485] link 'Reorder' url: link
          [3487] row ''
              [3488] cell '000000161'
              [2086] cell '2/27/23'
              [2087] cell '$762.18'
              [2088] cell 'Complete'
              [2089] cell 'View OrderReorder'
                  [3490] link 'View Order' url: link
                  [3492] link 'Reorder' url: link
          [3494] row ''
              [3495] cell '000000156'
              [2090] cell '2/24/23'
              [2091] cell '$231.54'
              [2092] cell 'Canceled'
              [2093] cell 'View OrderReorder'
                  [3497] link 'View Order' url: link
                  [3499] link 'Reorder' url: link
          [3501] row ''
              [3502] cell '000000158'
              [2094] cell '2/11/23'
              [2095] cell '$174.99'
              [2096] cell 'Canceled'
              [2097] cell 'View OrderReorder'
                  [3504] link 'View Order' url: link
                  [3506] link 'Reorder' url: link
          [3508] row ''
              [3509] cell '000000157'
              [2098] cell '2/9/23'
              [2099] cell '$185.32'
              [2100] cell 'Complete'
              [2101] cell 'View OrderReorder'
                  [3511] link 'View Order' url: link
                  [3513] link 'Reorder' url: link
      [4270] StaticText 'Items 1 to 10 of 37 total'
```

```
    [4272] StaticText 'Page'
    [2105] list 'Page'
        [3523] link 'Page 2' url: link
        [3527] link 'Page 3' url: link
        [3531] link 'Page 4' url: link
        [3535] link 'Page Next' url: link
    [4300] StaticText 'Show'
    [1986] combobox '' hasPopup: menu expanded: False
    [4303] StaticText 'per page'
    [4306] StaticText 'My Account'
    [3566] link 'My Account' url: link
    [4309] StaticText 'My Orders'
    [3570] link 'My Downloadable Products' url: link
    [3572] link 'My Wish List' url: link
    [3576] link 'Address Book' url: link
    [3578] link 'Account Information' url: link
    [3580] link 'Stored Payment Methods' url: link
    [3584] link 'My Product Reviews' url: link
    [3586] link 'Newsletter Subscriptions' url: link
    [3590] heading 'Compare Products'
    [4320] StaticText 'You have no items to compare.'
    [3595] heading 'Recently Ordered'
    [3596] generic 'Recently Ordered'
        [4324] StaticText 'Add to Cart'
        [2003] checkbox 'Add to Cart' checked: false
        [3604] link 'Plus Size Lingerie for Women Sexy for Sex Naughty Eyelash
                Lace Bodysuit Naughty Mesh One Piece Teddy Bodysuit Outfits' url: link
        [2008] checkbox 'Add to Cart' checked: false
        [3611] link 'Uttermost Volterra Crackled Taupe-Gray Ceramic Table Lamp' url: link
        [2009] checkbox 'Add to Cart' checked: false
        [3618] link 'NOZE Rustic Coat Rack Wall Mounted Shelf with 4 Hooks,
                Hanging Entryway Organizer for Mug Coffee Cup, Holding Solid Wooden
                Shelf with 2 Baskets for Kitchen Living Room, Bathroom
                and Bedroom' url: link
```

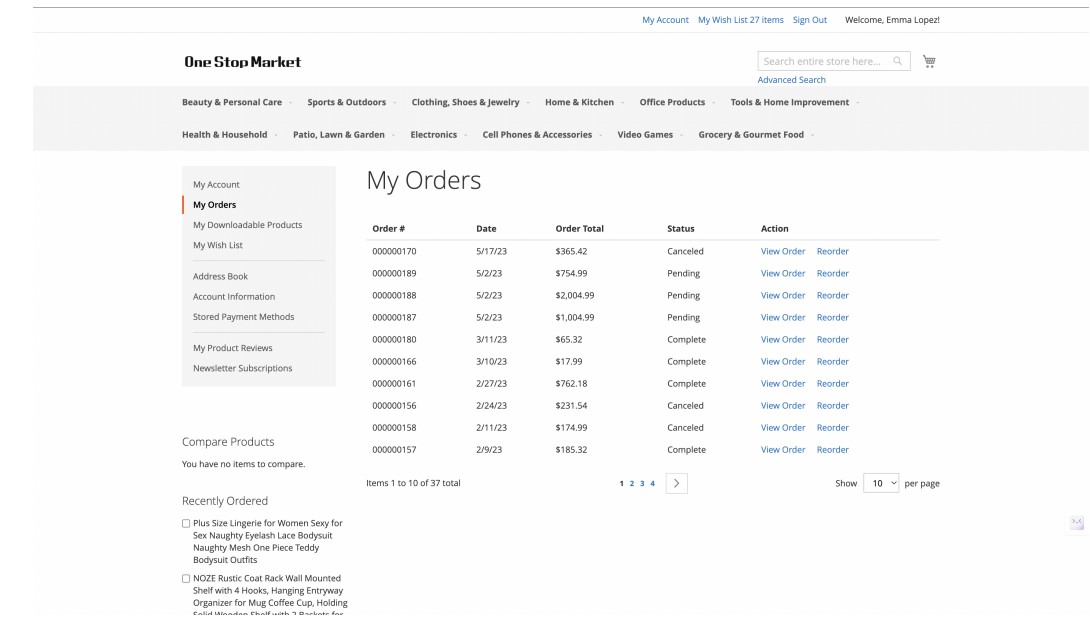

Figure 8: 'My Orders' Page of the User

```
Orchestrator Agent Dynamic Invocation – Planning Agent/Action Proposal Agent

"type": "spawn_agent", "next_agent_role": "Planning Agent", "reason": "The goal is
to get the total payment of the last completed orders. The current page displays a
table of orders, including their statuses and totals. To determine the optimal next
step (e.g., scrolling for more orders or selecting a specific order), we need candidate
actions that align with this goal. The Planning Agent will generate these action
candidates based on the current observation."
```

```
Prompt Adaptation for Planning Agent/Action Proposal Agent by Orchestrator Agent

You are the Planning Agent, responsible for determining the best action to take at the
current step in a web workflow automation task.
You work in collaboration with the Lookahead Agent, which provides an exploration of
possible actions along with their predicted consequences, pros and cons. However, your
role is to predict the best action that needs to be taken at the current step.

Given the current webpage content in the form of an AX Tree, the user's goal, and
the action history, your task is to analyze the available options and select the most
effective action that moves the workflow one step closer to the goal. The Lookahead
Agent provides navigation insights, such as predicted outcomes and goal alignment
scores for different actions, but your decision should also consider efficiency,
feasibility, and long-term workflow optimization.

Guidelines for Action Selection:
1. Understand the Context:
- Use the AX Tree to interpret the current webpage's structure and available elements.
- Refer to the user's goal and past actions to ensure logical continuity.
- Do not get stuck in a loop of actions where you are selecting the same action
repeatedly. The action history is there to help you understand the context and the
actions that have been taken so far. Try to avoid loops, and if you are in one, try to
break it by selecting a different action.

2. Determine the Best Action:
- Prioritize actions that minimize redundant steps and maximize progress toward the
goal.
- If multiple actions are equally viable, select the one that optimizes navigation
efficiency.
- Keep in mind that you can always navigate back to a previous page if you feel the
current page is not providing the necessary information to make a decision. This is an
important step to ensure that you are making an informed decision.

3. Justify the Selection:
- Clearly explain why the chosen action is the best option based on the goal, and the
action history.

Additional Context:
The user's goal is to get the total payment of the last completed orders. The current
page displays a table of past orders, with information such as order date, status
(including 'Completed'), and payment amounts. The current AX Tree indicates that the
page includes a scrollable list of order entries. The action history shows that the
agent has recently navigated to this order history page from a dashboard view, and has
not yet interacted with individual orders or scrolled through the list. The current
state suggests that further interaction|such as scrolling down to access more completed
orders|may be required to fulfill the goal. You should consider whether the visible
completed orders are sufficient to compute the total, or if additional data needs to be
revealed via scrolling or pagination.

Response Format (JSON): "actions": [ "action": "<Action 1>", "rationale":
"<Explain why Action 1 is a good next step>" , "action": "<Action 2>", "rationale":
"<Explain why Action 2 is a good next step>" , "action": "<Action 3>", "rationale":
"<Explain why Action 3 is a good next step>" ]
Do not output anything outside the JSON response, and strictly adhere to the format.
Do not add extra newlines or spaces in the JSON response format apart from the ones
already present in the prompt, so that it is easy to parse.
```

```
Response from the Planning/Action Proposal Agent

{ "actions": [ { "action": "click('2432', 'left')", "rationale": "This is a
potential action to display more completed orders, which could help in accessing the
fifth most recent order if it is not currently visible." }, { "action": "click('82',
'left')", "rationale": "This is a potential action to display additional completed
orders on the same page." }, { "action": "click('2227', 'left')", "rationale": "This
is a potential action to access details of an individual completed order. While not
necessary for computing the total from the list view, it may help confirm payment
information if needed." } ] }
```

```
Orchestrator Agent Dynamic Invocation - LookAhead Agent

{ "type": "spawn_agent", "next_agent_role": "Lookahead Agent", "reason": "As the
Planning Agent proposed three actions, the next step would be to evaluate whether each
of these actions lead to desirable next observation states. In order to forecast the
consequence of each action, we would spawn a Lookahead agent." }
```

```
Prompt Adaptation for LookAhead Agent by Orchestrator Agent

You are the Lookahead Agent, working in collaboration with a a suite of other agents to
assist in web workflow automation.  Your role is to evaluate the effectiveness of the
Planning Agent's actions by predicting the next page state and assessing their impact
on achieving the goal.  Also, you should analyse the possible future trajectory that is
possible considering the response of the Planning Agent in the current observation.

Given the current webpage content in the form of an AX Tree, the user's goal, and the
action history so far, your task is to identify and analyze the action predicted by
the Planning Agent.  Provide foresight on predicted next page states, potential future
trajectories.

**Guidelines for Lookahead**

1.  **Predict Immediate Consequences**
- Simulate the result of the action on the current webpage (e.g., navigation to a new
page, revealing hidden elements).
- If an action leads to dynamic content updates, specify what changes will occur.
- Look for repeated actions that the Planning Agent is getting stuck with by analysing
the action history and suggest a way to break the loop.

2.  **Look Ahead the Entire Workflow Trajectory**
- If a given action is taken, what are the next possible actions, and how should the
trajectory move ahead.
- Forecast how the state of the webpage evolves after subsequent interactions.

NOTE: Note that you have to lookahead the Planning Agent's actions based on the
predicted next page state, its alignment with the goal, and suggest the failure cases
possible out of the action taken by the Planning Agent.

The goal is to get the total payment amount of the last 5 completed orders.  Currently,
only 4 orders are visible in the "My Orders" section of the shopping website.  The
next action should focus on revealing the 5th order, which may require actions such as
scrolling, clicking certain links or any other button that displays more orders.

Now, consider the response provided by the Planning Agent, and based on the guidelines
provided, give your response by looking ahead in the possibilities of the navigation
trajectories.

Actions from the Planning Agent:  click('2432', 'left'), click('82', 'left'),
click('2227', 'left')

Response Format (JSON):
{ "actions":  [ { "action":  "<Action 1>", "lookahead":  "<Provide the lookahead for
Action 1>" }, { "action":  "<Action 2>", "lookahead":  "<Provide the lookahead for
Action 2>" }, { "action":  "<Action 3>", "lookahead":  "<Provide the lookahead for
Action 3>" } ] }

Do not output anything outside the JSON response, and strictly adhere to the format.
Do not add extra newlines or spaces in the JSON response format apart from the ones
already present in the prompt, so that it is easy to parse.
```

```
Response from the LookAhead Agent

{ "actions":  [ { "action":  "click('2432', 'left')", "lookahead":  "Navigating to the
next page may display additional orders, but it will result in the loss of visibility
for the currently listed 4 orders.  This would prevent obtaining the total payment
amount for the last 5 completed orders, making this action misaligned with the goal."
}, { "action":  "click('82', 'left')", "lookahead":  "This action is likely to increase
the number of displayed orders on the current page, allowing visibility of the 5th
order without losing access to the initially visible 4 orders.  This aligns well
with the goal of calculating the total payment for the last 5 completed orders." },
{ "action":  "click('2227', 'left')", "lookahead":  "Viewing an individual order may
provide detailed information, but it is redundant for calculating the total payment
since the payment amount is already visible in the list view.  Additionally, this
action will cause a loss of the list view context, making it difficult to access all 5
order amounts easily." } ] }
```

```
Orchestrator Agent for Dynamic Invocation/Final Action Prediction

{ "type":  "final_action", "final_action":  { "action":  "click('82', 'left')", },
"reason":  "In order to obtain the total payment of last of completed orders, the next
goal is to display the 5 completed order.  As per the forecast of the LookAhead agent,
the next optimal action is to click on show more button to reveal the 5th order." }
```

In this step, after navigating to '*My Orders*' page where the orchestrator initiates a deliberation between agents and invokes the action proposer agent which provides 3 candidate actions - clicking on *next page*; *show more*; or *view order details*. The orchestrator then invokes the lookahead agent and updates the prompt with the information that 4 orders are visible in current view. The lookahead agent then simulates the possibility that clicking on *next page* might result in loss of visibility of current 4 orders, and hence, favors clicking on *show more* which will likely show the 5th order. Therefore, the orchestrator finally select the show more option to reveal the 5th order. The baseline methods, instead chose to click on *next page* and got stuck in a loop.

### A.6.4 STEP 4: MY ORDERS' PAGE WITH 'SHOW MORE' CLICKED

```
Observation Space

[1988] RootWebArea 'My Orders' url: link
    [2351] link 'My Account' url: link
    [2353] link 'My Wish List 27 items' url: link
    [2355] link 'Sign Out' url: link
    [3794] StaticText 'Welcome, Emma Lopez!'
    [2360] link 'Skip to Content' url: link
    [2013] link 'store logo' url: link
        [1989] image 'one_stop_market_logo' url: link
    [2015] link 'My Cart' url: link
    [3811] StaticText 'Search'
    [1981] combobox 'Search' autocomplete: both hasPopup: listbox
            required: False expanded: False
    [2391] link 'Advanced Search' url: link
    [1997] button 'Search' disabled: True
    [2395] tablist '' multiselectable: False orientation: horizontal
        [2398] tabpanel ''
            [2019] menu '' orientation: vertical
                [2021] menuitem 'Beauty & Personal Care' hasPopup: menu
                [2024] menuitem 'Sports & Outdoors' hasPopup: menu
                [2027] menuitem 'Clothing, Shoes & Jewelry' hasPopup: menu
                [2030] menuitem 'Home & Kitchen' hasPopup: menu
                [2033] menuitem 'Office Products' hasPopup: menu
                [2036] menuitem 'Tools & Home Improvement' hasPopup: menu
                [2039] menuitem 'Health & Household' hasPopup: menu
                [2042] menuitem 'Patio, Lawn & Garden' hasPopup: menu
                [2045] menuitem 'Electronics' hasPopup: menu
                [2048] menuitem 'Cell Phones & Accessories' hasPopup: menu
                [2051] menuitem 'Video Games' hasPopup: menu
                [2054] menuitem 'Grocery & Gourmet Food' hasPopup: menu
    [3435] heading 'My Orders'
    [3440] table 'Orders'
        [3441] caption ''
        [3442] rowgroup ''
            [3443] row ''
                [3444] columnheader 'Order #' required: False
                [2059] columnheader 'Date' required: False
                [2060] columnheader 'Order Total' required: False
                [2061] columnheader 'Status' required: False
                [2062] columnheader 'Action' required: False
            [3446] row ''
                [3447] cell '000000170'
                [2063] cell '5/17/23'
                [2064] cell '$365.42'
                [2065] cell 'Canceled'
                [2066] cell 'View OrderReorder'
                    [3449] link 'View Order' url: link
                    [3451] link 'Reorder' url: link
            [3453] row ''
                [3454] cell '000000189'
                [2067] cell '5/2/23'
                [1991] cell '$754.99'
                [2068] cell 'Pending'
                [2069] cell 'View OrderReorder'
                    [3456] link 'View Order' url: link
                    [3458] link 'Reorder' url: link
            [3460] row ''
                [3461] cell '000000188'
                [2070] cell '5/2/23'
                [2071] cell '$2,004.99'
                [2072] cell 'Pending'
                [2073] cell 'View OrderReorder'
                    [3462] link 'View Order' url: link
                    [3464] link 'Reorder' url: link
            [3466] row ''
                [3467] cell '000000187'
```

```
            [2074] cell '5/2/23'
            [2075] cell '$1,004.99'
            [2076] cell 'Pending'
            [2077] cell 'View OrderReorder'
                [3469] link 'View Order' url: link
                [3471] link 'Reorder' url: link
        [3473] row ''
            [3474] cell '000000180'
            [2078] cell '3/11/23'
            [2079] cell '$65.32'
            [2080] cell 'Complete'
            [2081] cell 'View OrderReorder'
                [3476] link 'View Order' url: link
                [3478] link 'Reorder' url: link
        [3480] row ''
            [3481] cell '000000166'
            [2082] cell '3/10/23'
            [2083] cell '$17.99'
            [2084] cell 'Complete'
            [2085] cell 'View OrderReorder'
                [3483] link 'View Order' url: link
                [3485] link 'Reorder' url: link
        [3487] row ''
            [3488] cell '000000161'
            [2086] cell '2/27/23'
            [2087] cell '$762.18'
            [2088] cell 'Complete'
            [2089] cell 'View OrderReorder'
                [3490] link 'View Order' url: link
                [3492] link 'Reorder' url: link
        [3494] row ''
            [3495] cell '000000156'
            [2090] cell '2/24/23'
            [2091] cell '$231.54'
            [2092] cell 'Canceled'
            [2093] cell 'View OrderReorder'
                [3497] link 'View Order' url: link
                [3499] link 'Reorder' url: link
        [3501] row ''
            [3502] cell '000000158'
            [2094] cell '2/11/23'
            [2095] cell '$174.99'
            [2096] cell 'Canceled'
            [2097] cell 'View OrderReorder'
                [3504] link 'View Order' url: link
                [3506] link 'Reorder' url: link
        [3508] row ''
            [3509] cell '000000157'
            [2098] cell '2/9/23'
            [2099] cell '$185.32'
            [2100] cell 'Complete'
            [2101] cell 'View OrderReorder'
                [3511] link 'View Order' url: link
                [3513] link 'Reorder' url: link
    [4270] StaticText 'Items 1 to 10 of 37 total'
    [4272] StaticText 'Page'
    [2105] list 'Page'
        [3523] link 'Page 2' url: link
        [3527] link 'Page 3' url: link
        [3531] link 'Page 4' url: link
        [3535] link 'Page Next' url: link
    [4300] StaticText 'Show'
    [1986] combobox '' hasPopup: menu expanded: False
    [4303] StaticText 'per page'
    [4306] StaticText 'My Account'
    [3566] link 'My Account' url: link
    [4309] StaticText 'My Orders'
    [3570] link 'My Downloadable Products' url: link
    [3572] link 'My Wish List' url: link
    [3576] link 'Address Book' url: link
    [3578] link 'Account Information' url: link
    [3580] link 'Stored Payment Methods' url: link
    [3584] link 'My Product Reviews' url: link
    [3586] link 'Newsletter Subscriptions' url: link
    [3590] heading 'Compare Products'
    [4320] StaticText 'You have no items to compare.'
    [3595] heading 'Recently Ordered'
    [3596] generic 'Recently Ordered'
        [4324] StaticText 'Add to Cart'
        [2003] checkbox 'Add to Cart' checked: false
        [3604] link 'Plus Size Lingerie for Women Sexy for Sex Naughty Eyelash
            Lace Bodysuit Naughty Mesh One Piece Teddy Bodysuit Outfits' url: link
        [2008] checkbox 'Add to Cart' checked: false
```

```
        [3611] link 'Uttermost Volterra Crackled Taupe-Gray Ceramic Table Lamp' url: link
        [2009] checkbox 'Add to Cart' checked: false
        [3618] link 'NOZE Rustic Coat Rack Wall Mounted Shelf with 4 Hooks,
              Hanging Entryway Organizer for Mug Coffee Cup, Holding Solid Wooden Shelf
              with 2 Baskets for Kitchen Living Room, Bathroom and Bedroom' url: link
```

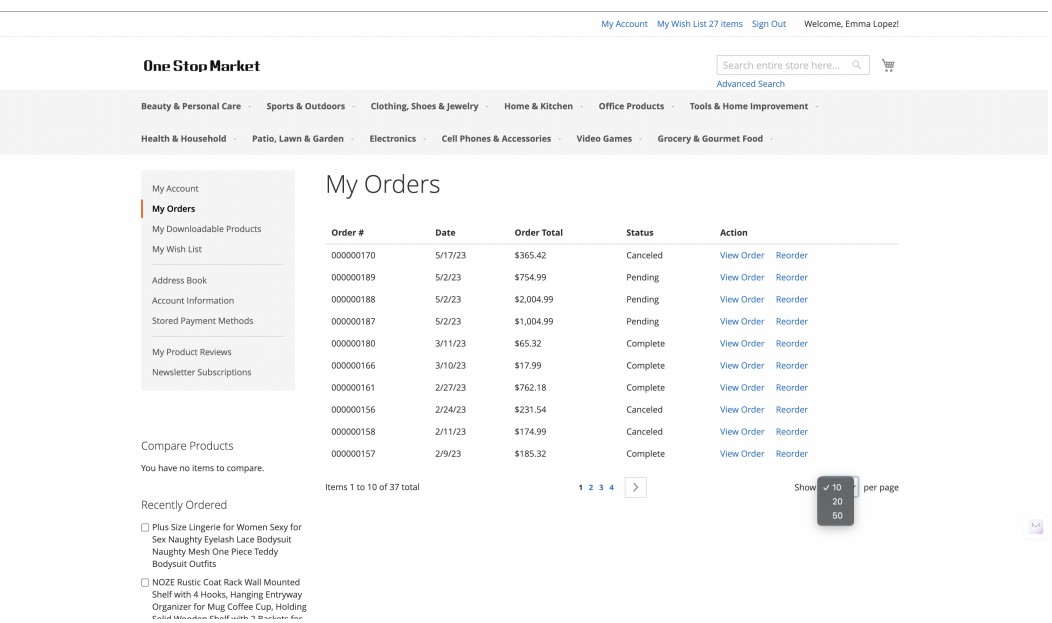

Figure 9: My Orders' Page with 'Show More' Clicked

```
Orchestrator Agent Dynamic Invocation - Memory Reflexion/Summarizer Agent

{ "type":  "spawn_agent", "next_agent_role":  "Memory Reflexion Agent", "reason":
"As the navigation trajectory has now exceeded four steps, it becomes increasingly
important to maintain a clear and concise understanding of the progress made so far.
To ensure effective decision-making and avoid redundant or suboptimal actions, I will
invoke the Memory Reflexion Agent to generate a summary of the interactions among all
previously spawned agents.  This summarization is especially valuable in the context
of the current goal | to compute the total amount of the last 5 completed orders | as
it will provide a high-level overview of the strategy followed so far, highlight any
bottlenecks or missed opportunities, and help guide future actions with better context
awareness." }
```

```
Prompt Adaptation of Memory Reflexion/Summarizer Agent by Orchestrator Agent

You are the Memory Reflexion Agent, working within a collaborative agent framework
designed for web workflow automation.  Your role is to simulate a deliberative memory
system|like a working memory loop|that helps the Orchestrator Agent reason more
effectively about next steps.

You analyze the recent trajectory of the task by reflecting on:
- The goal to be achieved
- The sequence of actions taken
- The observations (accessibility trees / page states) after each action

You distill this information into a compact memory trace that captures successes,
failures, patterns (such as repeated actions or loops), and potential bottlenecks
or progressions.  This memory serves as a short-term contextual window to support
decision-making by the Orchestrator Agent.

Input You Will Receive:
- **Goal:** The user's overarching intent
- **Action History:** Sequence of actions taken so far
- **Observation History:** Snapshots of page state (AX Trees) seen after each action

Guidelines for Memory Reflexion:
```

```
1.  **Compactly Summarize** the trajectory of steps taken, capturing critical events.
- What actions were taken and their immediate effects on state?

2.  **Highlight High-Impact Transitions**
- Identify whether any action led to significant progress or regression.
- Note if a particular pattern keeps repeating (e.g., looping clicks or returning to a
previous state).
- Detect any breakpoints where the workflow changed direction or got stuck.

3.  **Support Strategic Deliberation**
- State whether the workflow is converging towards the goal or drifting.

Goal:  <goal>

Action Space:  <action space>

Given below are the sequential webpage observations in the form of Accessibility
Trees along with the corresponding action history.  For each step, analyze the action
taken by the Planning Agent in the context of the current observation and the user's
goal.  Identify whether the action contributed positively toward progress or led to
redundancy, stagnation, or regression.  This analysis will guide the Planning Agent
in developing more effective strategies for navigating future states and achieving the
user's goal more efficiently.

Assume that the current observation is at step T, and the previous observations are at
steps T-1, T-2, T-3, and so on till T-k, with T-k being the first observation in the
navigation history.

Step T - {i}:
ACCESSIBILITY TREE:
ACTION HISTORY:

...
Response Format (JSON):
{ "memory_reflexion":  <Memory reflexion response about the navigation till now>, }

Do not output anything outside the JSON response, and strictly adhere to the format.
Do not add extra newlines or spaces in the JSON response format apart from the ones
already present in the prompt, so that it is easy to parse.
```

```
Response by Memory Reflexion/Summarizer Agent

{ "memory_reflexion":  "The goal is to find the total payment of the last five
completed orders.  The navigation began at the homepage of the shopping website, where
the Planning Agent was invoked to suggest possible next steps.  Among three recommended
actions, the optimal one|clicking on "My Account"|was chosen to progress toward
accessing the user's order history.  In Step 2, the user reached the "My Account"
page.  Based on the page structure, the Orchestrator Agent directly selected the action
to click on "My Orders" without invoking any additional agents, as it was an obvious
progression toward the goal.  In Step 3, the current observation was the "My Orders"
page, where the Planning Agent was again called.  It proposed three candidate actions:
clicking on the next page, clicking "Show More," or viewing individual order details.
Recognizing the complexity of choosing among these, the Orchestrator Agent then called
the Lookahead Agent, providing it with the context that only four orders were currently
visible.  The Lookahead Agent reasoned that clicking "Next Page" could cause the loss
of visibility of the current four orders, potentially hindering the calculation of the
last five payments.  It favored the "Show More" action as the safer and goal-aligned
choice.  Based on this recommendation, the Orchestrator selected the "Show More" action
to attempt to reveal the fifth order.  Notably, baseline methods in comparison had
taken the "Next Page" action and became stuck in a loop, highlighting the advantage of
the current agent-driven strategy.", }
```

```
Orchestrator Agent for Dynamic Invocation/Final Action Prediction

{ "type":  "final_action", "final_action":  { "action":  "send_msg_to_user('The total
payment amount for the last 5 completed orders is 1471.45$')", }, "reason":  "Given
the full contextual summary of the trajectory so far by the Memory Reflexion Agent, we
are currently in the 'My Orders' page with the last payment amount of 5 orders visible.
So, the total payment amount for the last 5 orders is 1471.45$" }
```

Based on the response of the Summarizer Agent, the Orchestrator itself decides to reveal the 5th completed order by clicking on displaying 20 orders in a single page.

### A.6.5  STEP 5: 'MY ORDERS' PAGE WITH LAST 5 COMPLETED ORDERS DISPLAYED

```
Observation Space

[2018] RootWebArea 'My Orders' url: link
    [2412] link 'My Account' url: link
    [2414] link 'My Wish List 27 items' url: link
    [2416] link 'Sign Out' url: link
    [3933] StaticText 'Welcome, Emma Lopez!'
    [2422] link 'Skip to Content' url: link
    [2049] link 'store logo' url: link
        [2019] image 'one_stop_market_logo' url: link
    [2051] link 'My Cart' url: link
    [3950] StaticText 'Search'
    [2022] combobox 'Search' autocomplete: both hasPopup: listbox
        required: False expanded: False
    [2453] link 'Advanced Search' url: link
    [2039] button 'Search' disabled: True
    [2457] tablist '' multiselectable: False orientation: horizontal
        [2460] tabpanel ''
            [2055] menu '' orientation: vertical
                [2057] menuitem 'Beauty & Personal Care' hasPopup: menu
                [2060] menuitem 'Sports & Outdoors' hasPopup: menu
                [2063] menuitem 'Clothing, Shoes & Jewelry' hasPopup: menu
                [2066] menuitem 'Home & Kitchen' hasPopup: menu
                [2069] menuitem 'Office Products' hasPopup: menu
                [2072] menuitem 'Tools & Home Improvement' hasPopup: menu
                [2075] menuitem 'Health & Household' hasPopup: menu
                [2078] menuitem 'Patio, Lawn & Garden' hasPopup: menu
                [2081] menuitem 'Electronics' hasPopup: menu
                [2084] menuitem 'Cell Phones & Accessories' hasPopup: menu
                [2087] menuitem 'Video Games' hasPopup: menu
                [2090] menuitem 'Grocery & Gourmet Food' hasPopup: menu
    [3497] heading 'My Orders'
    [3502] table 'Orders'
        [3503] caption ''
        [3504] rowgroup ''
            [3505] row ''
                [3506] columnheader 'Order #' required: False
                [2096] columnheader 'Date' required: False
                [2097] columnheader 'Order Total' required: False
                [2098] columnheader 'Status' required: False
                [2099] columnheader 'Action' required: False
            [3508] row ''
                [3509] cell '000000170'
                [2100] cell '5/17/23'
                [2101] cell '$365.42'
                [2102] cell 'Canceled'
                [2103] cell 'View OrderReorder'
                    [3511] link 'View Order' url: link
                    [3513] link 'Reorder' url: link
            [3515] row ''
                [3516] cell '000000189'
                [2104] cell '5/2/23'
                [2105] cell '$754.99'
                [2106] cell 'Pending'
                [2107] cell 'View OrderReorder'
                    [3518] link 'View Order' url: link
                    [3520] link 'Reorder' url: link
            [3522] row ''
                [3523] cell '000000188'
                [2108] cell '5/2/23'
                [2045] cell '$2,004.99'
                [2109] cell 'Pending'
                [2110] cell 'View OrderReorder'
                    [3525] link 'View Order' url: link
                    [3527] link 'Reorder' url: link
            [3529] row ''
                [3530] cell '000000187'
                [2111] cell '5/2/23'
                [2112] cell '$1,004.99'
                [2113] cell 'Pending'
                [2114] cell 'View OrderReorder'
                    [3532] link 'View Order' url: link
                        [3534] link 'Reorder' url: link
            [3536] row ''
                [3537] cell '000000180'
                [2115] cell '3/11/23'
                [2116] cell '$65.32'
                [2117] cell 'Complete'
                [2118] cell 'View OrderReorder'
                    [3539] link 'View Order' url: link
                    [3541] link 'Reorder' url: link
```

```
[3543] row ''
    [3544] cell '000000166'
    [2119] cell '3/10/23'
    [2120] cell '$17.99'
    [2121] cell 'Complete'
    [2122] cell 'View OrderReorder'
        [3546] link 'View Order' url: link
        [3548] link 'Reorder' url: link
[3550] row ''
    [3551] cell '000000161'
    [2123] cell '2/27/23'
    [2124] cell '$762.18'
    [2125] cell 'Complete'
    [2126] cell 'View OrderReorder'
        [3553] link 'View Order' url: link
        [3555] link 'Reorder' url: link
[3557] row ''
    [3558] cell '000000156'
    [2127] cell '2/24/23'
    [2128] cell '$231.54'
    [2129] cell 'Canceled'
    [2130] cell 'View OrderReorder'
        [3560] link 'View Order' url: link
        [3562] link 'Reorder' url: link
[3564] row ''
    [3565] cell '000000158'
    [2131] cell '2/11/23'
    [2132] cell '$174.99'
    [2133] cell 'Canceled'
    [2134] cell 'View OrderReorder'
        [3567] link 'View Order' url: link
        [3569] link 'Reorder' url: link
[3571] row ''
    [3572] cell '000000157'
    [2135] cell '2/9/23'
    [2136] cell '$185.32'
    [2137] cell 'Complete'
    [2138] cell 'View OrderReorder'
        [3574] link 'View Order' url: link
        [3576] link 'Reorder' url: link
[3578] row ''
    [3579] cell '000000148'
    [2139] cell '1/29/23'
    [2140] cell '$440.64'
    [2141] cell 'Complete'
    [2142] cell 'View OrderReorder'
        [3581] link 'View Order' url: link
        [3583] link 'Reorder' url: link
[3585] row ''
    [3586] cell '000000163'
    [2143] cell '1/16/23'
    [2144] cell '$132.24'
    [2145] cell 'Complete'
    [2146] cell 'View OrderReorder'
        [3588] link 'View Order' url: link
        [3590] link 'Reorder' url: link
[3592] row ''
    [3593] cell '000000154'
    [2147] cell '12/18/22'
    [2046] cell '$97.15'
    [2148] cell 'Complete'
    [2149] cell 'View OrderReorder'
        [3595] link 'View Order' url: link
        [3597] link 'Reorder' url: link
[3599] row ''
    [3600] cell '000000184'
    [2150] cell '12/14/22'
    [2151] cell '$20.49'
    [2152] cell 'Complete'
    [2153] cell 'View OrderReorder'
        [3602] link 'View Order' url: link
        [3604] link 'Reorder' url: link
[3606] row ''
    [3607] cell '000000162'
    [2154] cell '12/12/22'
    [2155] cell '$53.29'
    [2156] cell 'Complete'
    [2157] cell 'View OrderReorder'
        [3609] link 'View Order' url: link
        [3611] link 'Reorder' url: link
[3613] row ''
    [3614] cell '000000174'
    [2158] cell '12/4/22'
```

```
            [2159] cell '$32.47'
            [2160] cell 'Complete'
            [2161] cell 'View OrderReorder'
                [3616] link 'View Order' url: link
                [3618] link 'Reorder' url: link
        [3620] row ''
            [3621] cell '000000164'
            [2162] cell '11/26/22'
            [2163] cell '$218.17'
            [2164] cell 'Complete'
            [2165] cell 'View OrderReorder'
                [3623] link 'View Order' url: link
                [3625] link 'Reorder' url: link
        [3627] row ''
            [3628] cell '000000171'
            [2166] cell '11/20/22'
            [2167] cell '$133.07'
            [2168] cell 'Complete'
            [2169] cell 'View OrderReorder'
                [3630] link 'View Order' url: link
                [3632] link 'Reorder' url: link
        [3634] row ''
            [3635] cell '000000183'
            [3636] cell '11/11/22'
            [3637] cell '$51.94'
            [3639] cell 'Complete'
            [3640] cell 'View OrderReorder'
                [3641] link 'View Order' url: link
                [3643] link 'Reorder' url: link
        [3645] row ''
            [3646] cell '000000176'
            [3647] cell '10/22/22'
            [3648] cell '$845.07'
            [3650] cell 'Complete'
            [3651] cell 'View OrderReorder'
                [3652] link 'View Order' url: link
                [3654] link 'Reorder' url: link
    [4555] StaticText 'My Account'
    [3704] link 'My Account' url: link
    [4558] StaticText 'My Orders'
    [3708] link 'My Downloadable Products' url: link
    [3710] link 'My Wish List' url: link
    [3714] link 'Address Book' url: link
    [3716] link 'Account Information' url: link
    [3718] link 'Stored Payment Methods' url: link
    [3722] link 'My Product Reviews' url: link
    [3724] link 'Newsletter Subscriptions' url: link
    [3728] heading 'Compare Products'
    [4569] StaticText 'You have no items to compare.'
    [3733] heading 'Recently Ordered'
    [3734] generic 'Recently Ordered'
        [4573] StaticText 'Add to Cart'
        [2029] checkbox 'Add to Cart' checked: false
        [3742] link 'Plus Size Lingerie for Women Sexy for Sex Naughty Eyelash
            Lace Bodysuit Naughty Mesh One Piece Teddy Bodysuit Outfits' url: link
        [2034] checkbox 'Add to Cart' checked: false
        [3749] link 'NOZE Rustic Coat Rack Wall Mounted Shelf with 4 Hooks,
            Hanging Entryway Organizer for Mug Coffee Cup, Holding Solid Wooden Shelf
            with 2 Baskets for Kitchen Living Room, Bathroom and Bedroom' url: link
        [2035] checkbox 'Add to Cart' checked: false
        [3757] link 'Uttermost Volterra Crackled Taupe-Gray Ceramic Table Lamp' url: link
```

**Orchestrator Agent for Dynamic Invocation/Final Action Prediction**

{ "type": "final_action", "final_action": { "action": "send_msg_to_user('The total payment amount for the last 5 completed orders is 1471.45$')", }, "reason": "Given the full contextual summary of the trajectory so far by the Memory Reflexion Agent, we are currently in the 'My Orders' page with the last payment amount of 5 orders visible. So, the total payment amount for the last 5 orders is 1471.45$" }

In this step, since the page contains all the payment information of the last 5 completed orders, the orchestrator computed the total amount and sent it to the user.

AutoWeave demonstrates the power of dynamically invoking specialized agents—such as the Action Proposal Agent, Lookahead Agent, and Memory Reflexion Agent—based on evolving context and trajectory state. At each step, the Orchestrator Agent adapts the prompt intelligently, incorporating the user's goal, current observation, action history, and prior agent outputs to guide agent behavior effectively. This adaptive prompting ensures that each agent operates with the most relevant

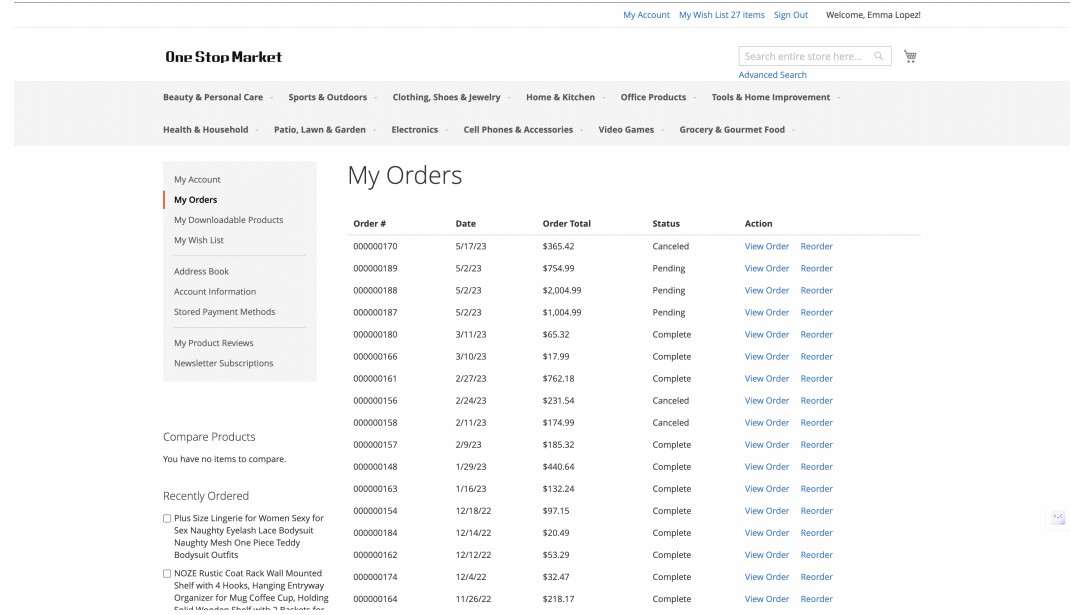

Figure 10: 'My Orders' Page with Last 5 Completed Orders Displayed

information, enhancing decision quality. As seen in the navigation toward computing the total payment of the last five completed orders, this collaborative agent architecture avoids common pitfalls (like getting stuck in loops) by promoting deliberate, goal-aligned action selection. The framework exemplifies how modular agent collaboration, contextual prompt adaptation, and strategic memory integration can enable robust, interpretable, and goal-driven web automation.

