# OpenReview forum: "AutoWeave: Automating Web Workflow Execution with Prompt-Adaptive Multi-Agent Orchestration"
_ICLR.cc/2026/Conference — Submitted to ICLR 2026_

### Official Review · Reviewer_6Ltc · 2025-10-30

**Soundness:** 2
**Presentation:** 2
**Contribution:** 3
**Rating:** 2
**Confidence:** 2

**Summary:**

The paper proposes an agentic framework for LLM-based web agents that anticipate future possibilities for step-wise action improvement with multi-agent orchestration. It also involves prompt refinement for adaptation to different sub-agent during hand-off. The proposed method is validated on WebVoyager and WebArena across multiple model families for generalization studies.

**Strengths:**

- The paper designs a suite of LLM agents which are targeted at various sub-tasks that might be encountered in the web scenarios. Such agent orchestration is comprehensive and adaptive to the context.
- The paper is generally intuitive and easy to follow, where each component and the workflow is explicitly described.

**Weaknesses:**

- The method appears to me somehow overfitting the benchmarks of WebVoyager and WebArena. All the designs of such workflows are not validated on OOD benchmarks (e.g., replace the observation reduction agent with other agent that can handle observations of different domains).
- The analysis of computing budget is not given for a fair comparison with baselines. It remains unknown the amount of tokens consumed for each baseline and how the proposed method performs on par with a similar resource budget. Especially from Fig.5, it can be observed that smaller turns (e.g., 5 turns) result in performance of WebArena (~23%) and WebVoyager (~36%) which are respectively lower than baselines (e.g., STeP, Agent-E).
- There exists no analysis on the performance of each role (sub-agent). For example, how well is the summarization? How accurate is the simulation of one-step forward? How effective is the prompt tuned for each task? Too many details are missing and honestly the reviewer cannot get much valuable information from the provided ablation study tables. How should we interpret the system? If anyone interested tries to apply the proposed method into their production scenario, how to measure the orchestration effectiveness?

**Questions:**

- Please consider in which part the proposed agent framework is unique to Web scenarios. And try to replace those components with general ones (e.g., sub-agent) for testing under OOD domains. Otherwise, the reviewer feels uncertain if the proposed framework can really make a difference rather than overfitting those two benchmarks.
- The comparison is not fair since the involved methods: 1) might not be reimplemented under the same experimental settings (instead directly cited only from their papers); 2) might consume different token budgets. The reviewer is concerned about the performance versus computing for comparison as scaling test-time computing is often positively associated with performance growth. It is unfair to simply run more turns (e.g., more reflection; more refinement; more voting) for performance gains.
- The illustration is poor with too small font size. Please consider improve the readability of texts in figures.
- Please add more discussions on the sub-agent performance itself. It is encouraged to present guidelines/take-home messages for readers on the design of each component.

---

### Official Review · Reviewer_iteB · 2025-11-01

**Soundness:** 3
**Presentation:** 3
**Contribution:** 3
**Rating:** 6
**Confidence:** 4

**Summary:**

This paper introduces AutoWeave. AutoWeave tackles a major limitation in web agents: fixed pipelines and static prompts cause stalls, over-deliberation, and broken behavior across diverse webpages and sub-tasks. The proposed system adds an LLM orchestrator that dynamically routes control among a few specialized agents with pre-defined roles (evidence reduction, action proposal, look-ahead, critic/validator, selector) and performs prompt adaptation each step to inject recent context and feedback. The key insight is that which agent to call next and how to prompt them matters as much as raw model capacity, as targeted agent routing plus small prompt edits reduces hallucinated actions and avoids wasteful tree search. Evaluation results on major web agent benchmarks (WebArena and WebVoyager) demonstrate that orchestration + prompt adaptation is a practical recipe for more reliable, cost-efficient web agents that remains modular and reproducible.

**Strengths:**

1. Very smart and solid design based on multi-agent and context adaptation philosophy. One thing I like about the design of AutoWeave is that the multi-agent workflow is not fixed. Instead, the orchestrator dynamically decide which specialized agent to call for the next step. After picking an agent, the orchestrator adapts its input prompt based on recent context and feedback. This is an idea widely adopted in recent work in agentic systems and context engineering, as this approach is training-free and flexible during runtime. Overall, the agentic system design is very reasonable and echoes with many classic design philosophy in multi-agent systems / context engineering.

2. Evaluation is comprehensive and results are good. Both WebArena and WebVoyager are very challenging and popular benchmarks in the research field. Numbers are really good as compared to existing web agent systems like STeP and AgentOccam (+6%~+10%). Microbenchmarks in 4.2, 4.3, and 4.4 show that AutoWeave is a generalizable and efficient solution.

3. Visualizations (Figures 1, 2, 3) are super helpful in terms of understanding how the agentic system works. As an example, after reading Figure 1, I feel that I can just skip section 3.1 --- I already have a crispy understanding of how different specialized agents work immediately.

**Weaknesses:**

1. For reproducibility and transparency, please consider releasing the prompts you use for the orchestrator, critic, lookahead, etc.

2. Evaluation lacks cost analysis, which is a major performance metrics of web agent workloads.

3. It'd be helpful to see some typical failure modes of AutoWeave --- Does most of the failures come from the orchestrator, e.g. not picking the right specialized agent? Or does most failures come from individual specialized agents?

**Questions:**

Please refer to "weaknesses".

One question regarding evaluation: On the official WebArena leaderboard, the number for STeP is 33.5, much higher than the one you have in table 1. Is this because your number and the leaderboard's number are based on different base LLMs?

---

### Official Review · Reviewer_YdWm · 2025-11-01

**Soundness:** 2
**Presentation:** 3
**Contribution:** 2
**Rating:** 4
**Confidence:** 2

**Summary:**

This paper presents AutoWeave, an agentic framework for automating web workflow execution using prompt-adaptive multi-agent orchestration. AutoWeave consists of a suite of LLM-based agents that deliberate to simulate action suitability and a dedicated Orchestrator agent that dynamically invokes the next appropriate agent and refines prompts based on workflow context. Evaluated on benchmarks WebVoyager and WebArena, AutoWeave achieves relative gains of 10% and 22% over the baselines respectively.

**Strengths:**

1. Clear Presentation: The paper is logically structured and easy to follow, with key concepts explained clearly to help readers understand the framework’s workflow and core value.
2. Comprehensive Agent Design: AutoWeave’s LLM-based agent suite covers critical web workflow automation needs, and the LookAhead Agent (functioning like a "world model" to simulate future states) addresses prior static frameworks’ shortcomings, with each agent’s role complementing others. The design of orchestrator agent is also intuitive.

**Weaknesses:**

1. Insufficient insightful Analysis. The paper reads more like an engineering report than a research paper, lacking in-depth analytical discussion. For example, it does not explain why the LookAhead Agent contributes significantly to performance (as shown in Table 4) or explore whether integrating context engineering methods (e.g., memagent [1]) into the Summarizer Agent could improve summarization quality.
2. Inadequate Comparison with Related Works on Agent Design: While the proposed agent suite is intuitive, the paper fails to clearly compare it with related works, specifically, which agents in the suite are unique to this study and how they differ from similar components in prior frameworks.

Reference:
[1] Yu, Hongli, et al. "MemAgent: Reshaping Long-Context LLM with Multi-Conv RL-based Memory Agent." arXiv preprint arXiv:2507.02259 (2025).

**Questions:**

See weaknesses. A key concern is the lack of insightful analysis and in-depth motivation for the agent suite. Each agent is directly proposed to fill a role, but the paper fails to sufficiently explain why each is necessary.

---

### Official Review · Reviewer_cfHJ · 2025-11-01

**Soundness:** 2
**Presentation:** 2
**Contribution:** 1
**Rating:** 2
**Confidence:** 4

**Summary:**

This paper proposes yet another web agent scaffolding that leverages (1) dynamic agent invocation: an orchestrator agent decides which agent to call next; (2) prompt adaptation: the orchestrator refines prompts contextually; (3) multi-agent deliberation: agents simulate possible future states (lookahead) and critique proposed actions before execution. The paper shows this method works across different model families (Llama, Qwen) on Webvoyager and WebArena.

**Strengths:**

1. Modularity: the agent suite (Observation Reduction, Action Proposal, LookAhead, Critic, Selector, etc) provides explicit functional decomposition. Figure 2 illustrates how deliberation happen and improves the clarity of the proposed workflow.
2. Efficiency analysis: discussion in section 4.4 is interesting. AutoWeave reduces redundant calls (186 LLM calls vs. >600 in Agent-E / >900 in Tree Search), showing its practical significance.

**Weaknesses:**

1. Limited novelty in underlying agent roles: While orchestration is novel, individual agents (summarizer, action reducer, lookahead, critic) largely repurpose existing LLM prompting paradigms. The contribution lies in combining them rather than designing new reasoning capabilities.
2. Lack of training: All components are prompt-engineered. No fine-tuning or self-improvement loop is explored.
3. Poor baseline selection: why not compare with newer methods on the WebArena leaderboard? The selected baselines are quite outdated.

**Questions:**

NA

---

### Meta-Review · Area_Chair_6jDh · 2026-01-07

**Summary:**

This paper develops an ochestrator model for enabling multiple LLms to check an appropriate next action in web navigation tasks. The reviewers commented about (i) lack of analysis, (ii) inadequate evaluation, e.g., checking different types of observations and tasks, and (ii) limited novelty. The authors did not provide a rebuttal. None of the reviewers strongly support the acceptance of the paper.

**Reviewer Concerns:**

The authors did not provide a rebuttal.

**Reviewer Scores:**

The scores provided by the reviewers reflect their comments adequately. Since there is no rebbutal from the authors, it is unclear whether the reviewers would have changed their scores after discussion.

---

### Decision · Program_Chairs · 2026-01-26

Reject